# Structural Inference: Interpreting Small Language Models with Susceptibilities

**Garrett Baker**$^{*=}$    **George Wang**$^{*=}$    **Jesse Hoogland**$^{*}$   **Vinayak Pathak**$^{*}$   **Daniel Murfet**$^{*}$

## Abstract

We develop a linear response framework for interpretability that treats a neural network as a Bayesian statistical mechanical system. A small perturbation of the data distribution, for example shifting the Pile toward GitHub or legal text, induces a first-order change in the posterior expectation of an observable localized on a chosen component of the network. The resulting *susceptibility* can be estimated efficiently with local SGLD samples and factorizes into signed, per-token contributions that serve as attribution scores. We combine these susceptibilities into a response matrix whose low-rank structure separates functional modules such as multigram and induction heads in a 3M-parameter transformer.

## 1 Introduction

The microscopic organization enabling the complex behaviors of neural networks remains poorly understood. This paper introduces *susceptibilities*, a novel interpretability framework rooted in statistical physics, to probe this internal structure. We treat a neural network as a Bayesian statistical mechanical system (Balasubramanian, 1997; LaMont and Wiggins, 2019) where an infinitesimal, controlled perturbation to the data distribution induces a first-order linear response in the expected behavior of a chosen network component, such as an attention head. This response, the susceptibility, quantifies the component's sensitivity to the specific data shift (Section 2) and is further related to geometry and generalization within singular learning theory Watanabe (2007).

**Contributions.**    Our main contribution is the development of a new interpretability paradigm derived from Bayesian learning theory and statistical physics. In particular:

- **We derive the theoretical framework of *susceptibilities*** for quantifying how model components respond to changes in the data distribution. This provides a principled link between structure in data and model internals.
- **We introduce the methodology of *structural inference*** for discovering internal structure in neural networks and attributing that structure to patterns in data. This yields new insight into how models balance expression and suppression.

Applying this methodology to a 3M-parameter transformer trained on the Pile (Section 4.2) we demonstrate that attention heads exhibit meaningfully differentiated susceptibilities to various data shifts. Our structural inference approach successfully distinguishes and separates known functional circuits from Hoogland et al. (2025),Wang et al. (2024) such as the induction circuit. This work thus provides a principled and empirically validated tool for dissecting the functional organization of neural networks, offering insights that align with and extend prior mechanistic studies.

## 2 Theory

### 2.1 Setup

We consider the model-truth-prior triplet $(p(y|x, w), q(x, y), \varphi(w))$ where $q(x, y) = q(y|x)q(x)$ is the true data-generating mechanism, $p(y|x, w)$ is the posited model of the conditional distribution

---

$^{*}$Timaeus
$^{=}$Equal contribution

with parameter $w \in W \subset \mathbb{R}^d$ representing the neural network weights, and $\varphi$ is a prior on $w$. We assume that $q(x, y) > 0$ and $p(y|x, w) > 0$ for all $(x, y) \in X \times Y$ and $w \in W$. We assume $W$ is compact. Let the sample spaces $X, Y$ be measure spaces so that $q(x, y)$ is a probability density with respect to the measure $\mu$ on $X \times Y$. In our intended application $X$ is the set of sequences of lengths $1 \le k < K$ in some alphabet $\Sigma$ of tokens and $Y = \Sigma$, with the counting measure.

Given a dataset $D_n = \{(x_i, y_i)\}_{i=1}^n$, drawn i.i.d. from $q(x, y)$, we define the sample negative log-likelihood function as $L_n(w) = -\frac{1}{n} \sum_{i=1}^n \log p(y_i|x_i, w)$ and its theoretical counterpart, the average negative log-likelihood or *population loss*, as $L(w) = -\mathbb{E}_{q(x,y)} \log p(y|x, w)$. The *annealed posterior* at inverse temperature $\beta > 0$ and sample size $n$ is

$$p_n^\beta(w) = \frac{1}{Z_n^\beta} \exp\{-n\beta L(w)\}\varphi(w) \quad \text{where} \quad Z_n^\beta = \int \exp\{-n\beta L(w)\}\varphi(w)\, dw. \tag{1}$$

## 2.2 VARIATION IN THE TRUTH

We consider a one-parameter variation of the data distribution of the following form. Let $H \subseteq \mathbb{R}$ be some open interval containing $0$ and for $h \in H$ let $q_h(x, y)$ be a probability density with respect to the measure $\mu$ on $X \times Y$. We assume that $h \mapsto q_h$ is differentiable as a map from $H$ to $L^1(X \times Y, \mu)$. We write $dxdy$ for $d\mu(x, y)$. The average negative log-likelihood for $q_h$ is $L^h(w) = -\mathbb{E}_{q_h(x,y)} \log p(y|x, w)$ and the annealed posterior is

$$p_n^\beta(w|h) = \frac{1}{Z_n^{\beta,h}} \exp\{-n\beta L^h(w)\}\varphi(w) \quad \text{where} \quad Z_n^{\beta,h} = \int \exp\{-n\beta L^h(w)\}\varphi(w)\, dw. \tag{2}$$

The simplest kind of one-parameter variation in the data distribution is a mixture. Let $q'$ be another probability density on $X \times Y$. Given $h \in [0, 1]$ we define

$$q_h = (1 - h)\, q + h\, q'. \tag{3}$$

Then the (one-sided) derivative exists and is the function $g_h(x, y) = q'(x, y) - q(x, y)$.

## 2.3 SUSCEPTIBILITIES

A *response function* measures how some observable changes under a small external perturbation or variation of a controlling parameter. In our Bayesian setting, suppose we pick an observable (a function or generalized function (Gelfand and Shilov, 1977) on parameter space) $\phi(w)$. We then consider its (annealed) posterior expectation, where we denote by $h$ some hyperparameter

$$\langle \phi \rangle_{\beta,h} = \int \phi(w) p_n^\beta(w|h) dw. \tag{4}$$

If we drop $h$ from the notation it means we are computing expectations with respect to the *unperturbed* (annealed) posterior (i.e. $h = 0$). A small change in $h$ induces a shift in the posterior as a distribution, and some aspect of this shift is captured by the shift in the expectation value $\langle \phi \rangle_{\beta,h}$.

**Definition 2.1.** The *susceptibility* of $\phi$ to the perturbation $q_h$ at inverse temperature $\beta$ is

$$\chi = \frac{1}{n\beta} \frac{\partial}{\partial h} \langle \phi \rangle_{\beta,h} \Big|_{h=0}. \tag{5}$$

**Lemma 2.2.** *The susceptibility for an observable $\phi$ is computed by*

$$\chi = -\operatorname{Cov}_\beta [\phi, \Delta L] \tag{6}$$

*where* $\operatorname{Cov}_\beta [\phi, \Delta L] = \langle \phi\, \Delta L \rangle_\beta - \langle \phi \rangle_\beta \langle \Delta L \rangle_\beta$ *and* $\Delta L = \frac{\partial L^h}{\partial h} \Big|_{h=0}$.

*Proof.* See Appendix A. □

**Lemma 2.3.** *For the (data-mixture) susceptibility, $q_h$ is the variation in (3), and we have $\Delta L(w) = L^1(w) - L(w)$ where $L^1(w)$ and $L(w)$ are $L^h(w)$ with $h = 1$ and $h = 0$, respectively.*

*Proof.* See Appendix A. □

From the lemma we learn that the susceptibility can be defined in terms of the unperturbed posterior. Next we define the primary class of observables to be considered in this paper. We consider models $p(y|x, w)$ parametrized by neural networks with weights $w \in W$. By a *component*, we mean a subset $C$ of these weights (e.g., an attention head in a transformer). Formally, a component is a product decomposition $W = U \times C$ with $U$ the parameters not in the component.

**Definition 2.4.** Given a component $C$ with $W = U \times C$ and a chosen parameter $w^* = (u^*, v^*)$ write $w = (u, v)$ and define

$$\phi_C(w) = \delta(u - u^*)\big[L(w) - L(w^*)\big] . \tag{7}$$

The motivation for using the observable $\phi_C$ in the setting of susceptibilities is that its expectation value is related to refined learning coefficients introduced in Wang et al. (2024). This means that we can think of the susceptibility associated to this observable as a kind of "rate of change" of the local learning coefficient, a geometric measure of model complexity (Appendix D.3). This provides a principled link between susceptibilities, generalization error in Bayesian statistics and changes in the geometry of the loss landscape with changes in the data distribution (Watanabe, 2009).

**Definition 2.5.** The *per-sample susceptibility* of $C$ for $(x, y) \in X \times Y$ is

$$\chi^C_{(x,y)} := -\mathrm{Cov}_\beta\big(\phi_C, \ell_{(x,y)} - L\big), \quad \ell_{(x,y)}(w) = -\log p(y|x, w) . \tag{8}$$

The notation is justified by the fact that the expectation of $\chi^C_{(x,y)}$ under the probe distribution $q'(x, y)$ is the susceptibility $\chi$ of $\phi_C$ (see Appendix A). In our applications $x$ is a sequence of tokens, $y$ is a single token and so we also refer to $\chi^C_{(x,y)}$ as the *per-token susceptibility*.

## 3 METHODOLOGY

In condensed matter physics, one way to define *internal structure* is that it is what causes the patterns that manifest in the responses of a system to external perturbations, as measured by natural observables (Altland and Simons, 2010, §7). In the previous section, we adapted this perspective to neural networks: changes in data are the external fields or perturbations we use to probe the system, changes in component-wise loss $\phi_C$ are the observables, and the susceptibility is the response.

In Section 3.1, we develop the *local susceptibility*, which enables us to practically estimate susceptibility matrices for individual neural network checkpoints. We then discuss how to interpret positive and negative susceptibility values as *suppression* and *expression*, respectively (Section 3.2). Finally, in Section 3.3, we introduce *structural inference*, a framework for identifying the patterns that manifest across a set of susceptibilities or, equivalently, for discovering internal structure.

### 3.1 ESTIMATING THE LOCAL SUSCEPTIBILITY

In practice, we are concerned with the properties of specific neural network checkpoints trained via stochastic optimizers, rather than ensembles drawn from the full Bayesian posterior. Moreover, sampling from global posterior is computationally infeasible. To circumvent these issues, we introduce *local* susceptibilities, which make it possible to estimate susceptibilities for individual models with samplers like Stochastic Gradient Langevin Dynamics (SGLD; Welling and Teh 2011).

We "localize" the posterior by replacing the prior $\varphi$ with a Gaussian prior centered at $w^*$, a local minimizer of $L(w)$. This ensures that sampling remains in a small neighborhood of $w^*$. We define a *local Gibbs posterior* (Bissiri et al., 2016) with inverse temperature $\beta > 0$,

$$p(w; w^*, \beta, \gamma) \propto \exp\left\{-n\beta L_n(w) - \frac{\gamma}{2}||w - w^*||_2^2\right\} , \tag{9}$$

as well as a *local annealed posterior*,

$$p(w; w^*, \beta, \gamma, h) \propto \exp\left\{-n\beta L^h(w) - \frac{\gamma}{2}||w - w^*||_2^2\right\} . \tag{10}$$

Given an observable $\phi$, we write the associated *local (annealed) expectation* as $\langle\phi; w^*\rangle_{\beta,h}$ which is the integral against this distribution $\int \phi(w)p(w; w^*, \beta, \gamma, h)dw$.

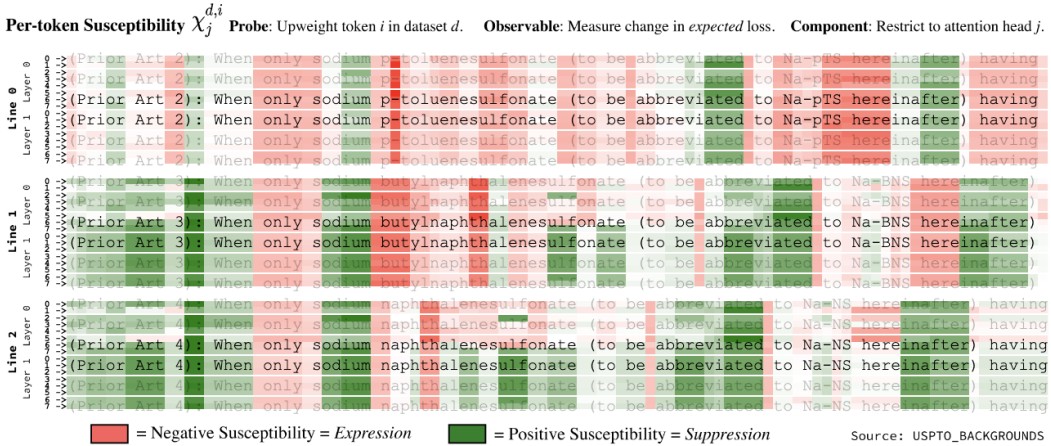

Figure 1: **Per-token susceptibilities reveal patterns of expression and suppression.** In this context, the beginning of three lines are shown. Each line is repeated six times, and each repeat is divided into three color bars. These bars correspond to individual heads 0:0-0:7 in layer 0 and 1:0-1:7 in layer 1, as shown on the left. The susceptibility values are standardized for each head, with solid red representing a z-score of -3 and solid green representing a z-score of +3. We see that the susceptibility increases for some heads on repeated token sequences like `Prior  Art`. These positive susceptibilities are examples of the suppression of induction patterns.

**Definition 3.1.** We define the *local susceptibility* at inverse temperature $\beta$ to be

$$\chi(w^*) = \frac{1}{n\beta} \frac{\partial}{\partial h} \langle \phi; w^* \rangle_{\beta,h} \bigg|_{h=0}. \tag{11}$$

By Lemma 2.2 we can write

$$\chi(w^*) = -\langle \phi \Delta L; w^* \rangle_\beta + \langle \phi; w^* \rangle_\beta \langle \Delta L; w^* \rangle_\beta. \tag{12}$$

We fix a mixture percentage $\delta h \in (0,1)$ and, as in Remark D.1, approximate $\Delta L$ by

$$\Delta L_n(w) := L_n^{\delta h}(w) - L_n(w) \tag{13}$$

for a dataset size $n$. We also replace the expectations in (12) over the annealed posterior (10) with expectations over the ordinary localized posterior (9) involving $L_n$. We obtain an estimator for the per-token susceptibility $\hat{\chi}_{(x,y)}$ by replacing $L_n^{\delta h}$ with $\ell_{(x,y)}$ (see Appendix C.4).

We compute $L_n(w)$ by drawing a sample from the original data distribution $q$, and $L_n^{\delta h}(w)$ by drawing a set of samples of size $n$, consisting of a mixture of samples from the unperturbed data distribution and samples from the perturbed data distribution (the mixture being controlled by $\delta h$). Given approximate samples $\{w_t\}_{t=1}^r$ from the (unperturbed and un-annealed) localized posterior with parameter $\beta$ we compute our estimate of the susceptibility to be

$$\hat{\chi}(w^*) := -\frac{1}{r} \sum_{t=1}^r \Big[ \phi(w_t) \Delta L_n(w_t) \Big] + \frac{1}{r^2} \Big[ \sum_{t=1}^r \phi(w_t) \Big] \Big[ \sum_{t=1}^r \Delta L_n(w_t) \Big]. \tag{14}$$

We further average this quantity over multiple chains $\{w_t\}_{t=1}^r$. For the precise expression that we implement in code in our experiments, see Appendix C.2.

## 3.2 INTERPRETING SUSCEPTIBILITIES AS *expression* AND *suppression*

The per-token susceptibility is defined as the first-order response of an expectation value to a change in the data distribution and as such, it is intrinsically meaningful. However there is a more concrete interpretation which is helpful in interpreting the empirical results in this paper.

Note that by linearity of the covariance

$$\chi^C_{(x,y)} = -\mathrm{Cov}_\beta\Big[\delta(u-u^*)\big[L(w)-L(w^*)\big], \ell_{(x,y)}(w)-L(w)\Big]$$

$$= \underbrace{-\mathrm{Cov}_\beta\Big[\delta(u-u^*)L(w),\ell_{(x,y)}(w)\Big]}_{\psi^C_{(x,y)}} + \underbrace{\mathrm{Cov}_\beta\Big[\delta(u-u^*)L(w),L(w)\Big]}_{\text{Does not depend on } x, y}$$

we get a decomposition of the per-token susceptibility into a part $\psi^C_{(x,y)}$ which depends on both $C$ and the token sequence $xy$ and a part that depends only on $C$. In our analysis we will form matrices of per-token susceptibilities and then standardize them by subtracting the mean across rows which are indexed by pairs $(x,y)$. This process of standardization cancels off the second term above and this means that our analysis ultimately depends only on the first summand $\psi^C_{(x,y)}$.

We call this the *standardized susceptibility* and now offer an interpretation of the sign of this term.

The standardized susceptibility measures how $\delta(u-u^*)L(w)$ and $\ell_{(x,y)}(w)$ covary when we perturb $w$ away from $w^*$, with perturbations being more likely according to their probability in the annealed posterior (that is, perturbations which increase the population loss $L(w)$ are exponentially suppressed). A variation in $w$ which only changes $L(w)$ by a small amount may nonetheless increase $\ell_{(x,y)}(w)$ for some tokens and lower it for others, and this correlation is what we care about.

**Negative standardized susceptibility** means that variations $w^* \to w$ which increase $\ell_{(x,y)}$ (that is, make $y$ less probable in context $x$) tend to be perturbations in the weights of $C$ which increase the loss overall. This makes sense if $(x,y)$ follows a pattern that $C$ is involved in predicting, mechanistically. Thus we associate negative susceptibility with *the component $C$ expressing that $y$ should follow $x$.*

**Positive standardized susceptibility** means that variations $w^* \to w$ which lower $\ell_{(x,y)}$ (that is, make $y$ more probable in context $x$) tend to be perturbations in the weights of $C$ which increase the loss overall. This makes sense if $(x,y)$ follows a pattern that $C$ is involved in "opposing", mechanistically. It could be predicting an alternative completion, or just decreasing the probability of this one. Thus we associate positive susceptibility with *the component $C$ suppressing the continuation of $x$ by $y$.*

| Sign of $\psi$ | | Interpretation |
|---|---|---|
| $\psi^C_{xy} < 0$ | Expression | Variations in $C$ which decrease loss, also raise $p(y|x,w)$. |
| $\psi^C_{xy} > 0$ | Suppression | Variations in $C$ which decrease loss, also lower $p(y|x,w)$. |

A visualization of the pattern of expression and suppression in natural language is given in Figure 1. For an analogy with magnetic susceptibility see Appendix D.1.

A natural question is how this susceptibility-centered notion of expression and suppression relates to existing work on prediction and suppression neurons in mechanistic interpretability, characterized through direct effects on logits (Gurnee et al., 2024b) (Lad et al., 2025). Our preliminary investigation does not reveal any simple relation (Appendix H).

### 3.3 SUSCEPTIBILITIES FOR STRUCTURAL INFERENCE

Our main application of susceptibilities is to discovering internal structure in neural networks. In our approach we start with two inputs: a finite set of components $\{C_j\}_{j\in\mathcal{H}}$ with associated observables $\phi_j := \phi_{C_j}$ (e.g. attention heads) and a finite set of data distributions $\{q^d\}_{d\in\mathcal{D}}$ with associated variations from $q$ by taking mixtures (e.g. Pile subsets like GITHUB). We refer to this as a *probe set* and the individual data distributions as *probe distributions*. The responses are by definition the entries of the $|\mathcal{D}| \times |\mathcal{H}|$ *data matrix* or *response matrix*

$$X = \big(\hat{\chi}^{C_j}_d\big)_{d\in\mathcal{D}, j\in\mathcal{H}} \tag{15}$$

where $\hat{\chi}^{C_j}_d$ is the estimated susceptibility of observable $\phi_{C_j}$ with respect to the variation of $q$ in the direction of $q^d$. Alternatively we can perform the analysis at the token level using variations in the data distribution which upweight continuations $y$ in contexts $x$, where some number of samples

$(x, y)$ are taken from each $q^d$ and we use a data matrix containing per-token susceptibilities for these samples (see Section 4.2 below).

In this view internal structure in the network means the linear algebraic structure of $X$ (to first-order). In Appendix D.4, we explain how for a particular formal definition of "patterns" in the data distribution (which we call modes, following Chen and Murfet (2025)) we can factor $X$ as a product $X = CP$ where $C$ consists of coupling constants between the probe distributions and modes, and $P$ couples observables and modes.

This motivates applying PCA to the data matrix and identifying the principal components (left singular vectors multiplied by singular values) with patterns in the data and the loadings (right singular vectors) with structures in the model. We call this approach, where we infer the internal structure in models via data analysis of matrices of susceptibilities, *structural inference*.

## 4 RESULTS

In this section, we apply our framework to study structure in a small language model. Our analysis automatically identifies attention heads' different functional roles, specializations, and higher-level structure like the "induction circuit." Following Elhage et al. (2021), Olsson et al. (2022), Hoogland et al. (2025) and Wang et al. (2024) we study two-layer attention-only (without MLP layers) transformers trained on next-token prediction on a subset of the Pile (Gao et al., 2020; Xie et al., 2023). For architecture and training details see Hoogland et al. (2025). Throughout, we refer to attention head $h \in \{0, \ldots, 7\}$ in layer $l \in \{0, 1\}$ by the notation $l{:}h$. The empirical loss for the transformer is defined in the usual way (see Appendix C.1).

We denote by $\Sigma$ the set of tokens. For tokenization, we used a truncated variant of the GPT-2 tokenizer that cut the original vocabulary of 50,000 tokens down to 5,000 (Eldan and Li, 2023). We denote token sequences as follows: `wa vel ength` is a sequence of three tokens. We set $X$ to be the disjoint union $X = \bigsqcup_{k=1}^{K-1} \Sigma^k$ and $Y = \Sigma$, both with the counting measure. We assume given a probability distribution $q_k(x, y)$ on $\Sigma^k \times \Sigma$ for $1 \leq k < K$ and define $q(x, y) = \frac{1}{K-1} q_k(x, y)$ for $x \in X, y \in Y$ with $x$ of length $k$. The conditional distribution $p(y|x, w)$ is parametrized by the transformer in the usual way (Phuong and Hutter, 2022).

### 4.1 SANITY CHECKS

We performed a range of checks to validate that our estimates of susceptibilities are non-trivial. While some of these are of independent interest, we have relegated them to appendices which we now summarize: (Appendix F) Across heads and datasets there is a significant variation in the susceptibilities (so there is signal); (Appendix E.1) The correlation of per-token susceptibilities with per-token losses and loss difference after ablation is very small (so the signal is not redundant), (Appendix E.2) Per-token susceptibilities tend to increase with context length especially for some layer 1 heads (so the signal depends also on $x$, not just $y$); (Appendix E.3) Per-token susceptibilities explain variation in overall susceptibilities (which justifies their use); (Appendix E.4) The same token $y$ can appear with positive or negative susceptibility depending on context.

### 4.2 FINDING INTERNAL STRUCTURE WITH SUSCEPTIBILITIES

We adopt the hypothesis from Wang et al. (2024) that internal structure in models reflects structure in the data distribution (Harris, 1954; Rogers and McClelland, 2004; Saxe et al., 2019). We therefore organize our results around six kinds of patterns, by which we mean a property of a token sequence $xy$ (full definitions in Appendix C.8; examples in Figure 2):

- **Word start:** a token that decodes to a space followed by lower or upper case letters.
- **Word part:** a non-word-end token that decodes to a sequence of upper or lower case letters.
- **Word end:** a token that decodes to a sequence of upper or lower case letters followed by a formatting token (see Appendix C.8 for an exhaustive list) delimiter or space.
- **Induction pattern:** a sequence of tokens $uvUuv$ where $U$ is a sequence of any length, $u, v$ are individual tokens, and $uv$ is not a common bigram, $q(v|u) \leq 0.05$.

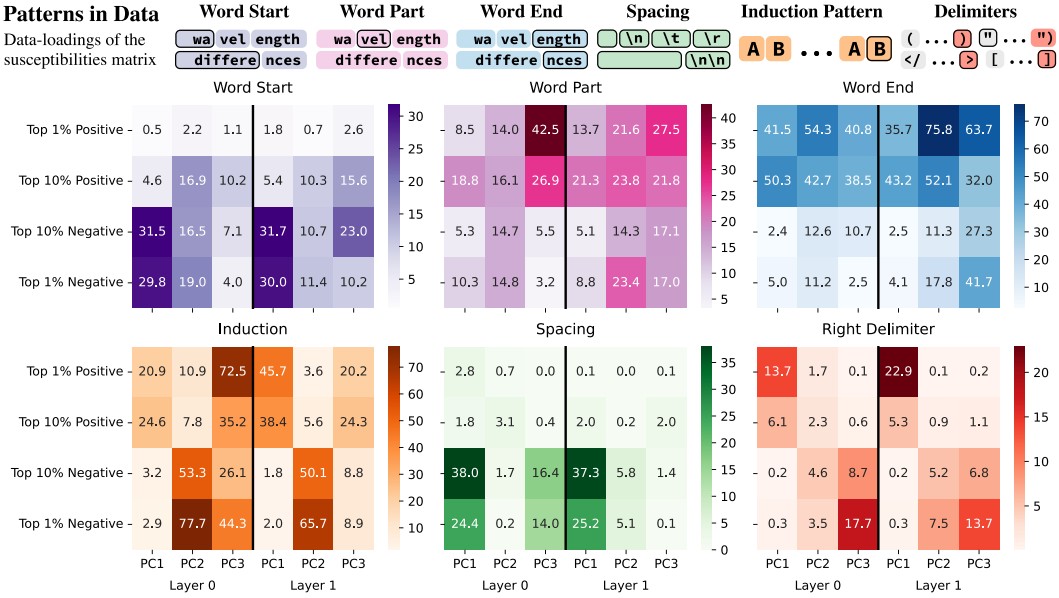

Figure 2: **Susceptibilities decompose into interpretable loadings over data.** Among the tokens with the coefficients of the largest magnitude in each principal component for the per-token susceptibility PCA, the percentage following each of the six patterns. The components are computed separately for each layer.

- **Spacing:** a token made up of one or more characters from ` `, `\n`, `\t`, `\r`, `\f`.
- **Right delimiter:** brackets and composite tokens, e.g. `)`, ` )`, `]`, `);`.

We randomly sample $N = 20000$ tokens from each dataset and form the data matrix

$$X(l) = (\hat{\chi}_{d,i}^{C_j})_{d \in \mathcal{D}, 1 \leq i \leq N, j \in \mathcal{H}(l)}$$

where $\mathcal{H}(l)$ indexes the heads in layer $l \in \{0, 1\}$, $\mathcal{D}$ all datasets, and $\hat{\chi}_{d,i}^{C_j} = \hat{\chi}_{(x,y)}^{C_j}$ is the estimated per-token susceptibility for the observable $\phi_{C_j}$ where $(x, y)$ is the $i$th sampled token $y$ in context $x$ in dataset $d$. The data matrix has size $N|\mathcal{D}| \times |\mathcal{H}(l)|$. We perform PCA on this data matrix (including standardizing the columns) and find the top three PCs explaining resp. $95.34\%, 1.83\%, 0.73\%$ of the variance for layer 0 and $99.14\%, 0.39\%, 0.11\%$ for layer 1. We examine the principal components to find the top positive and negative per-token susceptibilities and find the frequency of our six patterns among these top tokens. The resulting "data loadings" are shown in Figure 2, and in Appendix C.8 we include the background frequencies. The "component loadings" are shown in Figure 3. In the following we provide interpretations for each principal component.

**PC1: Word segmentation** The first PC is uniform across the heads in layer 0 and layer 1 and explains the majority of the variance. The interpretation is visually apparent in Figure 4: on most tokens the heads have similar susceptibilities which are broadly *positive* for word endings, induction patterns and right delimiters and *negative* for word starts and spaces. Note that, as with human infants, one of the first problems a language model must solve in acquiring language is word segmentation: identifying boundaries in a stream of tokens (Goldwater et al., 2009). It is therefore notable that the strongest pattern in the per-token susceptibilities seems to be associated with the model having learned to segment the token stream into words.

**PC2: The induction circuit** In data, PC2 is dominated by the dichotomy between word endings (positive) and induction patterns (negative). In components, PC2 has positive loadings on all heads identified in Wang et al. (2024) as being part of the induction circuit for this model, and negative loadings on the remaining heads. The *induction circuit* consists of the induction heads 1:6, 1:7 composing with the previous-token heads 0:1, 0:4 and current-token head 0:5.

In Wang et al. (2024) it was found that the negative effect of ablating the heads 1:0 - 1:5 was mainly on prediction of $n$-grams and skip $n$-grams and were thus termed layer-1 (L1) *multigram heads*. The

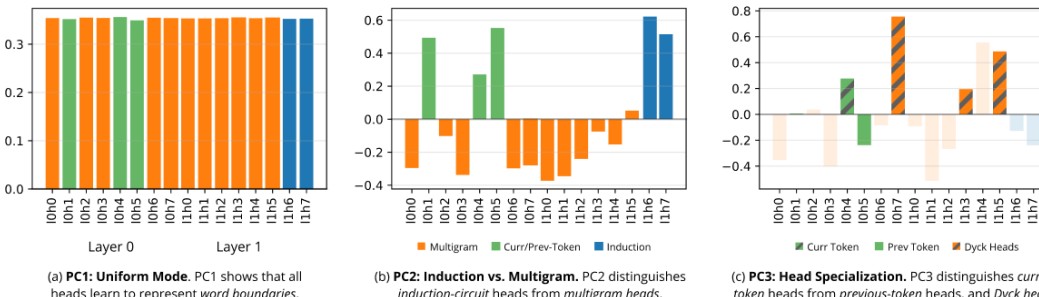

(a) **PC1: Uniform Mode**. PC1 shows that all heads learn to represent *word boundaries*.

(b) **PC2: Induction vs. Multigram.** PC2 distinguishes *induction-circuit* heads from *multigram heads*.

(c) **PC3: Head Specialization.** PC3 distinguishes *current-token* heads from *previous-token* heads, and *Dyck heads*.

Figure 3: **Susceptibilities decompose into interpretable loadings over components.** The loadings of the top two principal components for per-token susceptibility PCA on attention heads.

interpretation of PC2 is therefore that the induction circuit tends to express induction patterns which the remaining heads (including the layer 1 multigram heads) suppress, with the role of expression and suppression reversed for word endings.

We note that the appearance of the induction circuit as the heads with positive loadings in PC2 holds across all four independently trained seeds that we considered (Appendix I). In this sense *structural inference finds the induction circuit* and associates it with a pattern in the data distribution.

**PC3: Bracket matching** . In data, PC3 has more variation between the two layers than the earlier PCs, but in both layers there is a significant negative loading on right delimiters. This is interesting because among the five positively loaded components in Figure 3 are $\boxed{0:7}$, $\boxed{1:3}$, $\boxed{1:5}$ which were identified using ablations in Wang et al. (2024, Appendix B.3) as *Dyck heads* meaning that they are involved in predicting matching closing brackets of various kinds.

The role of $\boxed{1:4}$ here is unclear, but it was noted in (Wang et al., 2024, Appendix B.4.2) that this head seems to specialize in verb-particle phrases and other patterns including `://` ... `/` which could be viewed as generalized bracketings. The presence of this head in PC3 could also be explained by the data loadings on other patterns besides delimiters. Overall this component seems somewhat related to structure in the model for predicting brackets previously identified in Wang et al. (2024), although not as clearly as PC2 is related to the induction circuit.

To check the robustness of the claim to have distinguished the layer 1 multigram heads from the induction heads by per-token susceptibility PCA, we run an independent analysis in Appendix G.1 using PCA of the full susceptibilities including the values over training.

## 5 RELATED WORK

**Ablations and mechanistic interpretability.** Susceptibilities measure correlations of various changes in losses that result from perturbing the weights. A somewhat analogous set of techniques are ablations, which involve perturbing activations. These are widely used in mechanistic interpretability to test hypotheses about the computation being performed by parts of neural networks (Chan et al., 2022; Wang et al., 2023a; Rauker et al., 2023; Bereska and Gavves, 2024). By performing these interventions and observing the change in losses, researchers aim to infer facts about internal structure. In this paper we compare susceptibilities to zero ablation (Meyes et al., 2020; Hamblin et al., 2023; Nanda et al., 2023; Morcos et al., 2018; Zhou et al., 2018).

**Influence functions.** Classical influence functions measure the effect of up-weighting one training example on an estimator (Cook and Weisberg, 1980; Koh and Liang, 2017). Susceptibilities are closely related to the Bayesian form of influence functions (Giordano and Broderick, 2023; Iba, 2023). Suppose we have a dataset $D_n = \{(x_i, y_i)\}_{i=1}^{n}$. We introduce a single new sample $(x_{\text{new}}, y_{\text{new}})$ with negative log likelihood $\ell(w; x_{\text{new}}, y_{\text{new}})$ into the likelihood with a (small) inverse-temperature parameter $h$ controlling how "strongly" this new sample is weighted. Concretely, we define

$$p(w|D_n, h) \propto \exp\{-nL_n(w)\} \exp\{-h\,\ell(w; x_{\text{new}}, y_{\text{new}})\}\,\varphi(w).$$

Next we fix a test point $(x_{\text{test}}, y_{\text{test}})$ and define an observable $\phi(w) = \ell(w; x_{\text{test}}, y_{\text{test}})$. Then it is easy to check that $\chi$ in this case is $-\text{Cov}_{p(w|D_n)}\big[\ell(w; x_{\text{test}}, y_{\text{test}}), \ell(w; x_{\text{new}}, y_{\text{new}})\big]$. Thus

influence functions, in this sense, are a special case of susceptibilities. However, in this paper we focus on a different special case, where the observables are tied to parts of the model.

**Suppression behavior in neural networks.** Mechanistic interpretability has discovered many examples of suppression in language models, such as *anti-induction* heads that suppress induction patterns (Olsson et al., 2022; McDougall et al., 2024) and *negative name-mover* heads that decrease the probability of an earlier name being copied in the Indirect Object Identification (IOI) circuit (Wang et al., 2023a). McGrath et al. (2023) explore the phenomenon of *self-repair*, where later layers counteract the effects of ablation in earlier layers. McDougall et al. (2024) introduce the idea of *copy suppression* more generally, identifying a specific head in GPT2-small that actively inhibits copying patterns. Further evidence of copy suppression and self-repair across models in the Pythia family (Biderman et al., 2023) is presented in Tigges et al. (2024) and Rushing and Nanda (2024). Rushing and Nanda (2024) also speculate that self-repair might be an incorrect framing, since ablations are inherently evaluating the model off-distribution. These negative components are a challenge for scaling mechanistic interpretability (McDougall et al., 2024) as self-repair obscures the effect of ablations; see McGrath et al. (2023) and Rushing and Nanda (2024, §5.1).

Lad et al. (2024) suggest that predictions in neural networks with residual connections (such as transformers) result from an ensemble of many components which vote (Veit et al., 2016), and some of those votes may be "against". Gurnee et al. (2024a) study universal neurons across training seeds of GPT2 models and find consistently that models learn neurons in later layers that suppress particular sets of tokens. Building on a perspective in Geva et al. (2022) they endorse a view of networks building predictions by both "promoting and suppressing concepts in the vocabulary space". In this vein Yan and Jia (2025) show that language models answer factual queries by promoting many answers and then suppressing some of them. More generally, suppression or inhibition has been a key part of artificial neural networks since McCulloch and Pitts (1990); Wang et al. (2023b).

## 6 LIMITATIONS AND FUTURE WORK

The main limitation of our methodology is the quality of the approximate posterior sampling method. The effect of hyperparameter selection in SGLD on susceptibility estimation remains poorly understood. Furthermore, SGLD generates correlated samples. Averaging over multiple chains only partially mitigates this issue. In this paper we examine a small 3M parameter model, but as SGLD is inherently scalable, we do not expect major obstacles in applying the methods to larger models. For example, Wang et al. (2024) use SGLD to estimate similar observables in Pythia-70M. The main limitation for scalability is the need for separate posterior samples for each component; for details on costs up to the scale of 1.4B parameters see Appendix C.5.

## 7 CONCLUSION

We have introduced susceptibilities, a novel interpretability framework inspired by statistical physics. In this analogy we view a neural network as a *complex material* whose internal structure is reflected by the differential response of parts of that structure to a range of "external fields" in the form of variations in the data distribution. Applying this methodology to a 3M-parameter transformer demonstrates that this analogy is fruitful: attention heads exhibit meaningfully differentiated susceptibilities, and structural inference via response matrix analysis successfully separates known functional modules.

This work builds on the literature establishing singular learning theory (Watanabe, 2009) and local learning coefficient estimation as principled tools for understanding neural networks and their development (Lau et al., 2025; Hoogland et al., 2025; Wang et al., 2024; Urdshals and Urdshals, 2025; Carroll et al., 2025). The study of the balance of expression and suppression using susceptibilities fits naturally into existing literature within mechanistic interpretability (Section 5). We hope these techniques will lead to scalable and theoretically principled tools for interpretability in large neural networks, parallel to existing approaches within mechanistic interpretability such as ablations or sparse auto-encoders, but with deeper foundations in the mathematical theory of generalization.

## 8 REPRODUCIBILITY STATEMENT

Appendix C outlines experimental details of the model and of hyperparameters used in sampling and Appendix B lists hyperlinks to the datasets used.

## ACKNOWLEDGMENTS

This work was supported by the Survival and Flourishing Fund, Open Philanthropy (now Coefficient Giving), and the Manifund Regranting Program.

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

APPENDIX

The appendix provides supplementary material to support and expand upon the main text. It is organized as follows:

- **Appendix A:** Contains mathematical proofs for lemmas presented in the main text.
- **Appendix B:** Describes the datasets used in our experiments.
- **Appendix C:** This section details the experimental and computational methodologies, covering loss definitions (Appendix C.1), susceptibility estimation (Appendices C.2 to C.4), scalability to larger models (Appendix C.5), SGLD parameters (Appendix C.6), data mixing (Appendix C.7), and pattern definitions (Appendix C.8).
- **Appendix D:** This section offers further theoretical context for susceptibilities, including analogies to physics (Appendix D.1), discussions on perturbation scale (Appendix D.2), relation to Local Learning Coefficients (LLCs) (Appendix D.3), connection to data modes (Appendix D.4), and a geometric interpretation (Appendix D.6).
- **Appendix E:** This section presents additional empirical results, including metric comparisons (Appendix E.1), context length effects (Appendix E.2), explanations of susceptibility differences (Appendix E.3), bimodal token analysis (Appendix E.4), and further PCA details (Appendix E.5).
- **Appendix F:** Presents a detailed breakdown of top per-token susceptibilities across various attention heads and datasets.
- **Appendix G:** Details an alternative structural inference approach using PCA on susceptibility trajectories over training time.
- **Appendix I:** Analysis of the per-token susceptibility PCA for three additional training runs.
- **Appendix J:** Information on the sample variance of the principal components and loadings on heads.

# A PROOFS

*Proof of Lemma 2.2.* Let $Z_n^{\beta,h}$ be as in Section 2.2. Then

$$n\beta\chi = \frac{\partial}{\partial h}\langle\phi_h\rangle_{\beta,h}\Big|_{h=0} \tag{16}$$

$$= \frac{\partial}{\partial h}\left[\frac{1}{Z_n^{\beta,h}}\int\phi_h(w)\exp(-n\beta L^h(w))\varphi(w)dw\right]\Big|_{h=0} \tag{17}$$

$$= \frac{\partial}{\partial h}\frac{1}{Z_n^{\beta,h}}\Big|_{h=0}Z_n^\beta\langle\phi\rangle_\beta - n\beta\langle\phi\Delta L\rangle_\beta. \tag{18}$$

Since

$$\frac{\partial}{\partial h}\frac{1}{Z_n^{\beta,h}} = -\frac{1}{(Z_n^{\beta,h})^2}\int\left[-n\beta\frac{\partial}{\partial h}L^h(w)\right]\exp(-n\beta L^h(w))\varphi(w)dw \tag{19}$$

we have

$$\frac{\partial}{\partial h}\frac{1}{Z_n^{\beta,h}}\Big|_{h=0} = \frac{n\beta}{Z_n^\beta}\langle\Delta L\rangle_\beta. \tag{20}$$

Combining (18) and (20) we obtain

$$n\beta\chi = n\beta\langle\phi\rangle_\beta\langle\Delta L\rangle_\beta - n\beta\langle\phi\,\Delta L\rangle_\beta.$$

Dividing through by $n\beta$ gives the result. $\qquad\square$

*Proof of Lemma 2.3.* We compute

$$\begin{aligned}
\Delta L &= \frac{\partial}{\partial h}L^h\Big|_{h=0}\\
&= -\frac{\partial}{\partial h}\int q_h(x,y)\log p(y|x,w)dxdy\\
&= -\int\frac{\partial}{\partial h}q_h(x,y)\log p(y|x,w)dxdy\\
&= -\int(q'(x,y) - q(x,y))\log p(y|x,w)dxdy\\
&= L^1(w) - L(w)
\end{aligned}$$

using the definition of $\frac{\partial}{\partial h}q_h$ from Section 2.2. $\qquad\square$

**Lemma A.1.** *We have*

$$\chi = \int_{X\times Y}\chi_{(x,y)}\,q'(x,y)\,dx\,dy\,.$$

*Proof.* Exchanging the order of integration gives

$$\begin{aligned}
\int q'(x,y)\chi_{(x,y)}dx\,dy &= -\int q'(x,y)\,\mathrm{Cov}_\beta\left[\phi(w),\ell_{(x,y)}(w) - L(w)\right]dx\,dy\\
&= -\int q'(x,y)\Big\{\Big\langle\phi(w)\{\ell_{(x,y)}(w) - L(w)\};w^*\Big\rangle_\beta\\
&\qquad - \langle\phi(w);w^*\rangle_\beta\Big\langle\ell_{(x,y)}(w) - L(w);w^*\Big\rangle_\beta\Big\}dx\,dy\\
&= -\Big\langle\phi(w)\{L^1(w) - L(w)\};w^*\Big\rangle_\beta\\
&\qquad + \langle\phi(w);w^*\rangle_\beta\Big\langle L^1(w) - L(w);w^*\Big\rangle_\beta\\
&= -\mathrm{Cov}_\beta\left[\phi,\Delta L\right] = \chi
\end{aligned}$$

where $L^1(w) = \int q'(x,y)\ell_{(x,y)}(w)dx\,dy$. $\qquad\square$

Table 1: Details of the Pile (Gao et al., 2020) subset datasets used in our analysis.

| Dataset | Description | Size (Rows) |
|---|---|---|
| PILE-GITHUB | Code and documentation from GitHub | 100k |
| PILE-PILE-CC | Web crawl data from Common Crawl | 100k |
| PILE-PUBMED_ABSTRACTS | Scientific abstracts from PubMed | 100k |
| PILE-USPTO_BACKGROUNDS | Patent background sections | 100k |
| PILE-PUBMED_CENTRAL | Full-text scientific articles | 100k |
| PILE-STACKEXCHANGE | Questions and answers from tech forums | 100k |
| PILE-WIKIPEDIA_EN | English Wikipedia articles | 100k |
| PILE-FREELAW | Legal opinions and case law | 100k |
| PILE-ARXIV | Scientific papers from arXiv | 100k |
| PILE-DM_MATHEMATICS | Mathematics problems and solutions | 100k |
| PILE-ENRON_EMAILS | Corporate emails from Enron | 100k |
| PILE-HACKERNEWS | Tech discussions from Hacker News | 100k |
| PILE-NIH_EXPORTER | NIH grant applications | 100k |
| PILE_SUBSETS_MINI | Combined samples from all subsets | 6.66k |
| PILE1M | Combined samples from all subsets | 1M |

Note that in the main text we actually consider a mixture between $q$ and $(1 - \delta h)q + \delta h\, q'$ and so $\Delta L = L^{\delta h}(w) - L(w)$. Substituting in the above we obtain not $\chi = \int q'(x,y)\chi_{(x,y)}dx\,dy$ but rather

$$\chi = \delta h \int q'(x,y)\chi_{(x,y)}dx\,dy \tag{21}$$

since $L^{\delta h}(w) = (1 - \delta h)L(w) + \delta h\, L^1(w)$.

## B  DATA

The datasets we used to evaluate model performance are described in Table 1. These are generated by filtering for the first 100k rows from subsets of the uncopyrighted Pile (Devin Gulliver, 2025), which was generated by dropping copyrighted subsets from the Pile (Gao et al., 2020). Several subsets were omitted because there were too few rows to easily retrieve 100k samples.

We use two github datasets PILE-GITHUB and GITHUB-ALL. The former is a pile subset, while the latter is CODEPARROT/GITHUB-CODE and is included for consistency with (Wang et al., 2024).

## C  METHODS

### C.1  EMPIRICAL LOSS

Recall that $\Sigma$ denotes the set of tokens, and our sample space is $X \times Y$ where $X = \bigsqcup_{k=1}^{K-1} \Sigma^k$ and $Y = \Sigma$. For a probability distribution $P$ over tokens we denote by $P[t]$ the probability of token $t$, and $f_w$ is the function from contexts to probability distributions of next tokens computed by the transformer then for $x \in \Sigma^k$ and $y \in \Sigma$ we set

$$p(y|x, w) = \text{softmax}(f_w(x))[y].$$

See Phuong and Hutter (2022) for a formal definition of transformers. In practice for transformer training pairs $(x, y)$ are not sampled i.i.d, but instead are generating from *full contexts* $S_K = (t_1, \ldots, t_K) \in \Sigma^K$. We denote by $S_{\leq k}$ the sub-sequence $(t_1, \ldots, t_k)$ of $S_K$. Our dataset $D_n$ is a

---

All datasets are available on HuggingFace at `https://huggingface.co/collections/timaeus/datasets-pile-subsets-673a6a6c7ffc522a34ebfb0b`. These datasets are derived from The Pile (Gao et al., 2020) and specifically from the uncopyrighted version (Devin Gulliver, 2025), with cleaning performed by removing all content from the Books3, BookCorpus2, OpenSubtitles, YTSubtitles, and OWT2 subsets.

collection of full contexts, $\{S_K^i\}_{i=1}^n$. The *empirical loss* with respect to $D_n$ is

$$L_n(w) = -\frac{1}{n} \sum_{i=1}^n \frac{1}{K-1} \sum_{k=1}^{K-1} \log\left(\text{softmax}(f_w(S_{\leq k}^i))[t_{k+1}^i]\right). \tag{22}$$

We can think of this as the negative log-likelihood for biased samples $\{(S_{\leq k}^i, t_{k+1}^i)\}_{1 \leq k < K, 1 \leq i \leq n}$ from $q(x, y)$ over $X \times Y$ or as the negative log-likelihood for the unbiased samples $\{S_K^i\}_{i=1}^n$ over full contexts with probability model

$$p(S_K|w) = \prod_{k=1}^{K-1} p(t_{k+1}|t_1, \ldots, t_k, w).$$

In this paper $K = 1024$ and all full contexts begin with a BOS token `<|endoftext|>` (which has index 4999 in our tokenizer) so $p$ does not model the first token. In the case where we think of $p$ as modelling full contexts the factor of $\frac{1}{K-1}$ is unnatural (although standard in transformer training). If we set $l_n(w) = (K-1)L_n(w)$ then

$$n\beta L_n = n\frac{\beta}{K-1} l_n$$

so the effective inverse temperature in tempered posteriors is $\frac{\beta}{K-1}$. Similarly if we want to treat $L_n(w)$ as the empirical negative log-likelihood for the biased set of $n(K-1)$ samples then

$$n\beta L_n = n(K-1)\frac{\beta}{K-1} L_n$$

so the effective inverse temperature is again $\frac{\beta}{K-1}$.

## C.2 LOCAL SUSCEPTIBILITY

Let us consider the observable $\phi$ from (7) for some component $C$ and a variation in the data distribution as in (3) for some $q'$. Substituting this observable into (12), we denote the result by

$$\chi(w^*; C, q') = -\langle\phi\Delta L; w^*, C\rangle_\beta + \langle\phi; w^*, C\rangle_\beta \langle\Delta L; w^*\rangle_\beta \tag{23}$$

where $\langle -; w^*, C\rangle_\beta$ denotes an expectation defined as in Wang et al. (2024) where we fix the parameters $u = u^*$. To explain: the generalized function $\delta(u - u^*)$ in $\phi$ means that the expectations involving $\phi$ are computing an integral over the *weight-refined* posterior, where only parameters in the circuit $C$ (represented by the variables $v$) are allowed to vary. However the expectation $\langle\Delta L; w^*\rangle_\beta$ is defined with respect to the "full" posterior, where all parameters are allowed to vary.

If we let $\{w_t\}_{t=1}^r$ denote approximate samples from the weight-refined posterior for $C$ (that is, the same kind of samples we would use to define the *weight-refined* but *not* also data-refined LLC) and let $\{w_t'\}_{t=1}^r$ denote approximate samples from the full posterior (the kind of samples we would use to define the ordinary LLC) then the susceptibility is

$$\hat{\chi}(w^*; C, q') = -\frac{1}{r} \sum_{t=1}^r \left[\{L_n(w_t) - L_n(w^*)\}\Delta L_n(w_t)\right]$$
$$+ \frac{1}{r^2}\left[\sum_{t=1}^r \{L_n(w_t) - L_n(w^*)\}\right]\left[\sum_{t=1}^r \Delta L_n(w_t')\right]$$
$$= -\frac{1}{r} \sum_{t=1}^r \left[\{L_n(w_t) - L_n(w^*)\}\{L_n^{\delta h}(w_t) - L_n(w_t)\}\right]$$
$$+ \frac{1}{r^2}\left[\sum_{t=1}^r \{L_n(w_t) - L_n(w^*)\}\right]\left[\sum_{t=1}^r \{L_n^{\delta h}(w_t') - L_n(w_t')\}\right]. \tag{24}$$

## C.3 IMPLEMENTING SUSCEPTIBILITIES

Let $\hat{\chi}$ denote the susceptibility estimate (24) for the probe data distribution $q'$ and weight restriction $C$ where $w_t$ are SGLD weight samples from the weight restricted posterior, $w_t'$ are SGLD

weight samples from the full not weight restricted posterior, $w^\star$ are the initial model weights, $L_n$ is the loss function on the pretraining dataset $q$ on a random batch of size $n$, in this case $q = \texttt{timaeus/dsir-pile-1m-2}$, and $L_n^{\delta h}$ is the loss function on a random batch of size $n$ on $q_{\delta h} = (1 - \delta h)q + \delta h q'$. That is, a mixture between the pretraining dataset $q$ and probe dataset $q'$ using the process as specified in Appendix C.7.

These susceptibilities were found for each of the datasets listed in Appendix B, for each checkpoint step in

$$S = \{0, 100, 200, 300, 400, 500, 600, 700, 800, 900, 1000, 1500, 2000, 3000, 4000, 5000, 6000,$$
$$7000, 8000, 9000, 10000, 12500, 15000, 17500, 20000, 25000, 30000, 40000, 49900\}$$

and each attention head weight restriction, as well as no weight restriction at all. For these experiments we had a batch size of $n = 64$, and $\delta h = 0.1$. SGLD hyperprameters can be found in Appendix C.6.

The hardware used for the computation consisted of 10 A100 GPUs run in parallel, each computing all weight-refined susceptibilites on all datasets for an evenly distributed subset of $S$, each GPU took about 20 hours to complete its task.

### C.4 PER-TOKEN SUSCEPTIBILITIES

With the same notation as Appendix C.2 we estimate per-token susceptibilities using the equation

$$\hat{\chi}_{(x,y)} = -\frac{1}{r}\sum_{t=1}^{r}\left[\left\{L_n(w_t) - L_n(w^*)\right\}\left\{\ell_{(x,y)}(w_t) - L_n(w_t)\right\}\right]$$
$$+ \frac{1}{r^2}\left[\sum_{t=1}^{r}\left\{L_n(w_t) - L_n(w^*)\right\}\right]\left[\sum_{t=1}^{r}\left\{\ell_{(x,y)}(w_t') - L_n(w_t')\right\}\right].$$

For a total of approximately $M = 160 \times 1023 = 163680$ samples, across 160 different contexts, where each context $c^r = (c_1^r, \ldots, c_k^r)$ is a list of $k \leq 1023$ tokens, with the property that for all $j \in [2, k], (c_{1:j-1}^r, c_j^r) \in D_M^1$.

We choose these contexts using the $\texttt{datasets.Dataset.shuffle}$ method with seed 0 after tokenizing the dataset under consideration, and taking the 160 contexts.

To visualize these per-sample susceptibilities, let

$$\hat{\chi}(c_j^r) = -\frac{1}{r}\sum_{t=1}^{r}\left[\left\{L_n(w_t) - L_n(w^*)\right\}\left\{\ell_{(c_{1:j-1}^r, c_j^r) - L_n(w_t)}(w_t)\right\}\right]$$
$$+ \frac{1}{r^2}\left[\sum_{t=1}^{r}\left\{L_n(w_t) - L_n(w^*)\right\}\right]\left[\sum_{t=1}^{r}\left\{\ell_{(c_{1:j-1}^r, c_j^r) - L_n(w_t')}(w_t')\right\}\right].$$

Then this is a real number, and we can map it to a color value via either or where the only differences

---

**Require:** $\hat{\chi}(c_j^r) \in \mathbb{R}$
1: $\max(\hat{\chi}) \leftarrow \max_{r,i}|\hat{\chi}(c_j^r)|$
2: **if** $\hat{\chi}(c_j^r) > 0$ **then**
3:     color $\leftarrow \texttt{rgba}\left(0, 255, 0, \frac{|\hat{\chi}(c_j^r)|}{\max(\hat{\chi})}^2\right)$
4: **else**
5:     color $\leftarrow \texttt{rgba}\left(255, 0, 0, \frac{|\hat{\chi}(c_j^r)|}{\max(\hat{\chi})}^2\right)$
6: **end if**

---

are the shade of green and the quadratic versus linear opacity scaling.

Each $c_j^r$ can also be decoded into a string of characters using our model's tokenizer. Therefore we can highlight each such string the corresponding color computed above. Let $s(c_j^r)$ be this highlighted

---

**Require:** $\hat{\chi}(c_j^r) \in \mathbb{R}$
1: $\max(\hat{\chi}) \leftarrow \max_{r,i} |\hat{\chi}(c_j^r)|$
2: **if** $\hat{\chi}(c_j^r) > 0$ **then**
3:      color $\leftarrow$ rgba $\left(0, 128, 0, \frac{|\hat{\chi}(c_j^r)|}{\max(\hat{\chi})}\right)$
4: **else**
5:      color $\leftarrow$ rgba $\left(255, 0, 0, \frac{|\hat{\chi}(c_j^r)|}{\max(\hat{\chi})}\right)$
6: **end if**

---

string. Then this produces Figure 9 using the first coloring algorithm while the latter is used to produce Figure 4 and Figure 1.

For Figure 9, we sort each per-sample susceptibility in a descending manner, giving us the list $\hat{\chi}(c_{j_1}^{r_1}) \geq \cdots \geq \hat{\chi}(c_{j_M}^{r_M})$. We then choose the first 100 of these susceptibilities, and visualize their surrounding context by concatenating each of their highlighted strings. With $+$ denoting concatenation,

$$
\begin{aligned}
s(c_{j_1-200}^{r_1}) \quad + \quad \cdots \quad + \quad s(c_{j_1+200}^{r_1}) \\
\vdots \\
s(c_{j_1-200}^{r_{100}}) \quad + \quad \cdots \quad + \quad s(c_{j_1+200}^{r_{100}})
\end{aligned}
$$

And we also choose the last 100 of these susceptibilities, and visualize the surrounding context in the same way

$$
\begin{aligned}
s(c_{j_1-200}^{r_M}) \quad + \quad \cdots \quad + \quad s(c_{j_1+200}^{r_M}) \\
\vdots \\
s(c_{j_1-200}^{r_{M-100}}) \quad + \quad \cdots \quad + \quad s(c_{j_1+200}^{r_{M-100}})
\end{aligned}
$$

For Figure 4 and Figure 1, we use the same context samples from each dataset but do no sorting of the data and instead search manually for representative examples.

Here we use all the same definitions and parameters as in Appendix C.3, and SGLD sampling with hyperparameters as in Appendix C.6, modified to only use 100 draws. We run these per-token susceptibilities for all datasets in Appendix B and on checkpoint step 49900.

The hardware used for this computation consisted of 16 A100 GPUs, each computing a quarter of the weight restrictions for each seed. Each GPU took 50 hours to complete its task

### C.5 SCALABILITY TO LARGER MODELS

There are two main obstacles to scaling the results in the paper:

1. The computational cost of SGLD sampling and evaluating losses to compute susceptibilities.
2. The complexity of larger models.

Regarding the first point, we estimate the cost of compute as follows: for a given model, let $T$ be the time cost of a training step, $F$ be the cost of a forward pass, $k$ the number of probe datasets, $m$ the number of model components being analyzed, and $C$ some roughly constant O(100) number of SGLD samples. Then we expect that the cost of estimating susceptibilities on a given checkpoint of that model scales like

$$
mC(T + kF).
$$

In practice, we take approximately log-spaced checkpoints of models, so the total cost of analysis scales logarithmically with the total number of training steps and typically ends up comparable to the cost of training. Some preliminary work suggests that we may be able to significantly reduce the computational cost in larger models without loss of signal, such as by using much less than the entire set of attention heads. Not accounting for these cost improvements, we were for example able to run susceptibilities on the full set of attention heads (384 in total) on Pythia 410m and Pythia 1.4b

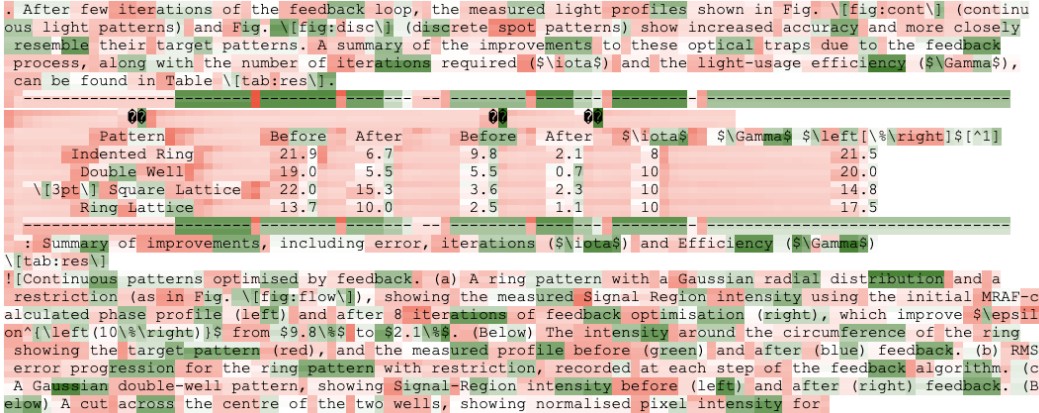

Figure 4: **Visualizing per-token susceptibilities for three heads on a sample from ARXIV.** Each token is highlighted in three segments (top, middle, buttom) which correspond to the per-token susceptibilities for three heads (0:0, 1:2, 1:7). The susceptibility values are standardized for each head, with solid red representing a z-score of -3 and solid green representing a z-score of +3. Green means positive susceptibility and red means negative.

(Biderman et al., 2023) for around $2,000 and $5,000 respectively, for a single checkpoint each. For sampling more generally, Urdshals et al. (2025) establishes LLC estimation at scale by studying Pythia models up to 6.9B (see e.g. Figure 3).

We also note that, apart from SGLD sampling, the scaling behavior of the compute cost is similar to e.g. circuit discovery via ablations, which also requires evaluating forward passes for a set of model components being ablated.

As model size grows, these computational costs require paralellization across multiple GPUs, but this is done in the standard way and is well-supported. Overall, there are no theoretical or practical opstables to scaling to 10B parameters, and the costs are not prohibitive.

Regarding (2), the difficulties are more uncertain. The small transformers studied in the present paper are simple, and so the patterns in the data and corresponding structures in the model are remarkably visible in the susceptibility matrix. We do not expect PCA on its own to be sufficient to discover interesting structure in larger models, which "understand" more complex patterns. In these settings we also make use of non-linear dimension reduction techniques like UMAP, and more sophisticated data analysis to study the internal structure in models. The present paper should be viewed as validation that simple data analysis, in simple models, reveals simple patterns; we agree that it remains an open question (to be addressed in future work) whether more complex data analysis reveals complex structures in larger models.

Intuitively, our method should be thought of as somewhat comparable to EK-FAC for data attribution: while it is computationally nontrivial to estimate Hessian inverses in large models and the results are not straightforward to interpret, this is routinely done (see e.g. R. Grosse et al "Studying large language model generalization with influence functions") because the information gained is believed to be worth the cost.

## C.6    SGLD HYPERPARAMETERS

Following Wang et al. (2024) we used SGLD with hyperparameters $\gamma = 300$, $n\beta = 30$, $\varepsilon = 0.001$, batch size 64, 4 chains, and 200 draws for the regular susceptibilities as in Appendix C.3, but only 100 draws and batch size 16 for the per-token susceptibilities as in Appendix C.4.

Note that for these experiments we did not use RMSProp SGLD.

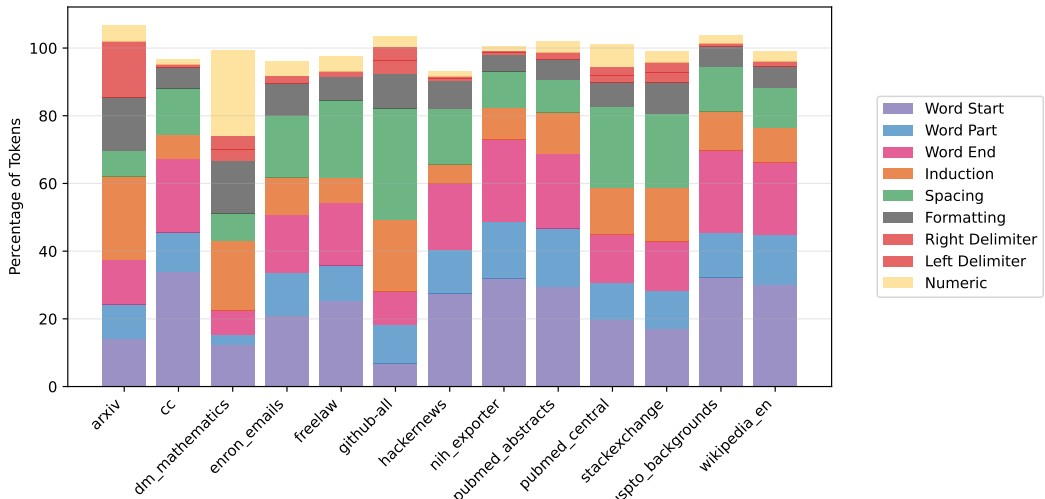

Figure 5: Percentages of tokens in each dataset which follow a given pattern. Note that not all patterns are mutually exclusive, which is why some totals exceed 100.

### C.7 DATASET MIXING

To create the mixed dataset $D^{\delta h}$ consisting of samples from $q_{\delta h} = (1 - \delta h)q + \delta h q'$, we interleaved the pretraining dataset with each probe dataset using Algorithm 1 in order to ensure even mixing, determinacy, and the property that if $q' = q$, then $D^{\delta h} = D^0$.

Let $(x_i, y_i)$ be samples from $q$ and $(x'_i, y'_i)$ be samples from $q'$, with $N = 1,000,000$ the size of the pretraining dataset, and $M = 100,000$ the size of the probe dataset. PILE1M

---

**Algorithm 1** Data Mixing Algorithm

---

**Require:** $D_N^0 = \{(x_i, y_i)\}_{i=1}^N, D_M^1 = \{(x'_i, y'_i)\}_{i=1}^M, \delta h \in [0, 1]$
1: $D_{\min(N,M)}^{\delta h} \leftarrow \emptyset$
2: probe count $\leftarrow 0$
3: $j \leftarrow 1$
4: **while** $j \leq N, M$ **do**
5:     target probe count $\leftarrow \lfloor (j + 1)\delta h \rfloor$
6:     **if** probe count $<$ target probe count **then**
7:         $D_{\min(N,M)}^{\delta h} \leftarrow D_{\min(N,M)}^{\delta h} \cup \{(x'_j, y'_j)\}$
8:         probe count $\leftarrow$ probe count $+ 1$
9:     **else**
10:         $D_{\min(N,M)}^{\delta h} \leftarrow D_{\min(N,M)}^{\delta h} \cup \{(x_j, y_j)\}$
11:     **end if**
12:     $j \leftarrow j + 1$
13: **end while**

---

This new dataset is then shuffled, and during SGLD minibatches of size 64 are drawn from this shuffled pre-computed data-distribution for loss estimates.

### C.8 TOKEN PATTERNS

Let $\Sigma$ denote the set of tokens in our tokenizer.

**Definition C.1.** A *left delimiter token* is an element of the set of tokens

$$\left\{ \text{<,} \quad \text{<,} \quad \text{\{,} \quad \text{\{,} \quad \text{(,} \quad \text{(,} \quad \text{[,} \quad \text{[,} \quad \text{</,} \quad \text{\{",} \quad \text{\$} \right\}.$$

A *right delimiter token* is an element of the set of tokens

$$\left\{ \ \texttt{>} \ , \ \texttt{>} \ , \ \texttt{\}} \ , \ \texttt{\}} \ , \ \texttt{)} \ , \ \texttt{)} \ , \ \texttt{]} \ , \ \texttt{]} \ , \ \texttt{),} \ , \ \texttt{],} \ , \ \texttt{):} \ , \ \texttt{).} \ , \ \texttt{))} \ , \ \texttt{);} \ , \ \texttt{\%)} \ , \ \texttt{\$} \ \right\}.$$

We call a token a *delimiter token* if it is either a left or right delimiter token.

The asymmetry between left and right delimiters is due to the tokenizer. For our model right delimiters seem much more important than left delimiters.

**Definition C.2.** A *formatting token* is an element of the set of tokens

$$\Big\{ \ \texttt{\~{}} \ , \ \texttt{\textbackslash\textbackslash} \ , \ \texttt{\textbackslash\textbackslash} \ , \ \texttt{/} \ , \ \texttt{//} \ , \ \texttt{//} \ , \ \texttt{://} \ , \ \texttt{-} \ , \ \texttt{-} \ , \ \texttt{--} \ , \ \texttt{--} \ , \ \texttt{\_} \ ,$$

$$\texttt{========} \ , \ \texttt{--} \ , \ \texttt{----} \ , \ \texttt{--------} \ , \ \texttt{----------------} \ , \ \texttt{**} \ , \ \texttt{****} \ , \ \texttt{********} \ ,$$

$$\texttt{\#\#\#\#} \ , \ \texttt{.} \ , \ \texttt{,} \ , \ \texttt{:} \ , \ \texttt{::} \ , \ \texttt{:} \ , \ \texttt{;} \ , \ \texttt{;} \ , \ \texttt{",} \ , \ \texttt{<|endoftext|>} \ ,$$

$$\texttt{="} \ , \ \texttt{":"} \ , \ \texttt{|} \ , \ \texttt{'} \ , \ \texttt{"} \ , \ \texttt{->} \ , \ \texttt{->} \ , \ \texttt{\^{}} \ , \ \texttt{\%} \ \Big\}.$$

**Definition C.3.** A *word start* is a single token that begins with a space and is followed by lower or upper case letters. That is, it is a token which when de-tokenized matches the regular expression `" [A-Za-z]+$"`.

**Definition C.4.** A *spacing token* is a token which when de-tokenized is a sequence of characters from the set

$$\left\{ \ \ , \ \texttt{\textbackslash n} \ , \ \texttt{\textbackslash t} \ , \ \texttt{\textbackslash r} \ , \ \texttt{\textbackslash f} \ \right\}.$$

**Definition C.5.** A *numeric token* is a token which when de-tokenized and with spaces removed, consists of one or more digits.

The patterns defined above are independent of the context in which a token appears. By contrast, the subsequent definitions apply to a token in a given context.

**Definition C.6.** A *word end token* is a token which when de-tokenized is made up of upper or lower case letters and which is followed in its context by a single formatting token, delimiter or space.

**Definition C.7.** A *word part token* is a token which is not a word ending in its context and which when de-tokenized consists of upper or lower case letters.

**Definition C.8.** An *induction pattern* is a sequence $xyUxy$ where $U \in \Sigma^*$ and $x, y \in \Sigma$, satisfying the following conditions:

- The conditional probability of $y$ following $x$ satisfies $q(y|x) \le 0.05$.

- $x, y \notin \{ \ \ , \ \texttt{\textbackslash n} \ , \ \texttt{,} \ , \ \texttt{.} \ , \ \texttt{the} \ , \ \texttt{to} \ , \ \texttt{:} \ , \ \texttt{and} \ , \ \texttt{by} \ , \ \texttt{in} \ , \ \texttt{a} \ , \ \texttt{be} \ \}$.

Note that $U$ can be the empty sequence and may contain occurrences of $x, y$. In a given context we classify a token as an *induction pattern token* if it is $y$ for an induction pattern $xyUxy$ within the context.

We use estimated conditional probabilities based on samples from the Pile. Note that the sets of left delimiters, right delimiters, formatting tokens, word start tokens and word part tokens are pairwise disjoint. The set of induction pattern tokens and word part tokens are disjoint. The percentage of $N = 20000$ tokens sampled from each dataset which fit each of these patterns are given in Figure 5.

# D  THEORETICAL BACKGROUND

## D.1  ANALOGY WITH MAGNETIC SUSCEPTIBILITY

The Bayesian susceptibility introduced above is the direct analogue of susceptibilities in physics, such as the magnetic susceptibility. We recall briefly the linear response of a spin system placed in an external magnetic field $H$. With energy $E(\sigma)$ (which may also depend on $H$) and magnetization $M(\sigma) = \sum_i \sigma_i$, the field-perturbed Boltzmann weight is

$$p_H(\sigma) = \frac{1}{Z_H} \exp\big[-\beta\big(E(\sigma) - H\,M(\sigma)\big)\big], \quad Z_H = \int \exp[-\beta(E - HM)]\,d\sigma.$$

The *magnetic susceptibility* is the first derivative of the equilibrium magnetisation,

$$\chi_{\mathrm{mag}} = \frac{\partial}{\partial H} \langle M \rangle_H \Big|_{H=0}, \quad \langle M \rangle_H = \int M(\sigma)\, p_H(\sigma)\, d\sigma \,.$$

Its sign carries immediate physical meaning:

| Sign of $\chi_{\mathrm{mag}}$ | | Interpretation |
|---|---|---|
| $\chi_{\mathrm{mag}} > 0$ | (paramagnetism) | spins align with $H$; response *reinforces* the field |
| $\chi_{\mathrm{mag}} = 0$ | (insensitive) | system insensitive or already saturated |
| $\chi_{\mathrm{mag}} < 0$ | (diamagnetism) | induced currents oppose $H$; response *cancels* the field |

The Bayesian susceptibility $\chi$ parallels the magnetic susceptibility, with the probe parameter $h$ replacing the magnetic field strength $H$ and $\phi(w)$ replacing the magnetization $M(\sigma)$.

### D.2 THE SCALE OF $h$

Since the scale of the parameter $h$ is arbitrary, there is a sense in which susceptibilities are only well-defined up to a rescaling. This is easy to understand if we consider multiple variations of the form (3), say $q \to q'$ and $q \to q''$ where the KL divergence between $q, q''$ is much larger than between $q, q'$. By default therefore we should not read too much into comparisons of *magnitudes* of susceptibilities across different variations of the data distribution.

For example if we set $\bar{h} = \frac{h}{\kappa}$ for some $\kappa > 0$ then

$$\frac{1}{n\beta} \frac{\partial}{\partial \bar{h}} \langle \phi \rangle_{\beta,\bar{h}} \Big|_{\bar{h}=0} = \frac{\kappa}{n\beta} \frac{\partial}{\partial h} \langle \phi \rangle_{\beta,h} \Big|_{h=0} = \kappa\, \chi \,. \tag{25}$$

It would also be natural to include a factor $1/\delta h$ in (13) so that $\Delta L_n(w)$ is a finite difference quotient. This could be motivated by the principle given above that we should rescale the deformation parameter $h$ since our interval is "really" of length $\delta h$.

*Remark* D.1. Given a target distribution $q'(x, y)$ and the mixture (3) we obtain $\Delta L = L^1(w) - L(w)$ from Lemma 2.3. However, if the difference between $q'$ and $q$ is large, $w^*$ may not be a local minima of $L^1(w)$. The likelihood of this being a reasonable assumption increases as we move $q'$ closer to $q$.

Since susceptibilities are in any case infinitesimal, this suggests that we replace the original variation from $q$ to $q'$ with a variation from $q$ to $(1 - \delta h)q + \delta h q'$ for some small $\delta h$. In this case the mixture is

$$h \in [0, 1] \longmapsto (1 - h)q + h(1 - \delta h)q + h\delta h q'$$
$$= (1 - h\delta h)q + h\delta h q' \,.$$

In this case, again by Lemma 2.3, we will have that $\Delta L$ is a finite difference

$$\Delta L(w) := L^{\delta h}(w) - L(w) \,.$$

For local susceptibilities we always choose a variation which is sufficiently small, in this sense.

### D.3 SUSCEPTIBILITIES AS A DERIVATIVE

Let $w^*$ be a local minima of $L(w)$. Recall from Lau et al. (2025); Watanabe (2009) that the estimated local learning coefficient (LLC) is defined by

$$\hat{\lambda}(w^*) = n\beta \left[ \mathbb{E}^{\beta}_{w|w^*,\gamma}[L_n(w)] - L_n(w^*) \right], \tag{26}$$

where $\mathbb{E}^{\beta}_{w|w^*,\gamma}$ is the expectation with respect to the Gibbs posterior (9).

Suppose we approximate the derivative in the local susceptibility by a difference quotient

$$\chi_{\mathrm{finite}}(w^*) := \frac{1}{n\beta} \frac{1}{\delta h} \left[ \langle \phi; w^* \rangle_{\beta,\delta h} - \langle \phi; w^* \rangle_{\beta} \right] \,.$$

When the observable $\phi$ is associated to a component $C$ as in Theorem 2.4 but with a dependence on $h$

$$\phi(w) = \delta(u - u^*)\Big[L^h(w) - L^h(w^*)\Big]$$

the expectations in question are, up to scalars, estimates of the weight- and data-refined LLCs as introduced in Wang et al. (2024) once we replace the annealed posteriors by the ordinary ones, since

$$\mathbb{E}^\beta_{w|w^*,\gamma}\Big[\delta(u - u^*)\{L_n^h(w) - L_n^h(w^*)\}\Big] = \frac{1}{n\beta}\hat{\lambda}(w^*; q_h, C)\,.$$

Hence $\chi_{\text{finite}}(w^*)$ is related to the difference quotient

$$\frac{1}{(n\beta)^2}\frac{1}{\delta h}\Big[\hat{\lambda}(w^*; q_{\delta h}, C) - \hat{\lambda}(w^*; q, C)\Big]$$

which measures a rate of change of the refined LLC as a function of the shift in the data distribution.

## D.4 Modes and matrix factorizations

In the main text we have focused on a decomposition of susceptibilities $\chi$ as an integral of *per-token* susceptibilities $\chi_{(x,y)}$ (Theorem A.1). Such a presentation implicitly relates to a choice of token sequences as a basis of a function space containing the conditional distributions $q(y|x)$, and it is conceptually useful to consider an alternative basis of "patterns" or more precisely *modes*. In this appendix we explain how to make this precise in the setting of (Chen and Murfet, 2025).

That paper explains how to think about the data distribution over sequences as a tensor, and use tensor decomposition methods to produce natural orthonormal bases of function space. Similar techniques are commonly used in the field of natural language processing (Anandkumar et al., 2012).

We fix a finite alphabet of tokens $\Sigma$ and let

$$\mathscr{H} = \mathscr{H}_{k,l} = L^2(\Sigma^k, q_k; \mathbb{R}^{\Sigma^l})$$

be the space of functions $\Sigma^k \to \mathbb{R}^{\Sigma^l}$ where $k, l > 0$. In the main text $1 \le k < K - 1$ and $l = 1$. Here $\mathbb{R}^{\Sigma^l}$ denotes the vector space of formal linear combinations of sequences of tokens of length $l$. The inner product is defined by

$$\langle f, g \rangle = \int \langle f(x), g(x) \rangle_{\mathbb{R}^{\Sigma^l}} q(x)dx\,.$$

We set $X = \Sigma^k, Y = \Sigma^l$ with the counting measure, $q(x, y)$ is the distribution for $x \in \Sigma^k, y \in \Sigma^l$ and $q(x) = q_k(x)$ denotes the distribution over sequences of length $k$. Let $\mathcal{C} \in \mathscr{H}$ be the function corresponding to the true conditional probabilities

$$\mathcal{C}(x) = \int q(y|x)\, y dy\,, \qquad \forall x \in \Sigma^k\,. \tag{27}$$

As explained in Chen and Murfet (2025) by applying tensor decomposition methods such as SVD to $\mathcal{C}$ we obtain a natural orthonormal basis $\{e_{\alpha\beta}\}_{\alpha\in\Lambda, \beta\in\Lambda^{++}}$ of $\mathscr{H}$. Here $\Lambda$ indexes right singular vectors, $\Lambda^+$ indexes left singular vectors for nonzero singular values, and $\Lambda^{++} \supseteq \Lambda^+$ is an extended set of indices associated to an arbitrary extension of the left singular vectors to an orthonormal basis of $\mathbb{R}^{\Sigma^l}$.[*] The examples given in Chen and Murfet (2025) justify relating such modes to *atomic patterns* in the data distribution. In this section when we sum over $\alpha, \beta$ we always mean $\alpha \in \Lambda$ and $\beta \in \Lambda^{++}$ respectively. Sometimes we use $\gamma$ in place of $\beta$.

Let us consider a variation of the data distribution $q(x, y)$ which is a "concept shift", that is, we vary the conditional distribution $q(y|x)$ but not $q(x)$. Let us write $q'(y|x)$ for some other conditional distribution and set $q'(x, y) = q'(y|x)q(x)$. Let $\mathcal{C}'$ denote the equivalent of (27) but for $q'$.

Recall that $\ell_{(x,y)}(w) = -\log p(y|x, w)$. Let $\chi$ be a susceptibility as defined in Section 2.3.

---

[*]For consistency with Chen and Murfet (2025) we continue to use $\beta$ as an index, note this is distinct from the inverse temperature.

**Definition D.2.** We call

$$\mathcal{C}' - \mathcal{C} = \sum_{\alpha,\beta} c^{\alpha\beta} e_{\alpha\beta} \qquad (28)$$

the *mode decomposition* of the data distribution shift.

**Definition D.3.** Given $w \in W$ we define $\Phi(w) \in \mathscr{H}$ by

$$\Phi(w)(x) = \int \ell_{(x,y)}(w) \, y dy \, .$$

Given $\alpha \in \Lambda$ we define $\Phi_{\alpha\beta} : W \longrightarrow \mathbb{R}$ by $\Phi_{\alpha\beta}(w) = \langle \Phi(w), e_{\alpha\beta} \rangle_{\mathscr{H}}$.

**Lemma D.4.** *We have*

$$\chi = -\sum_{\alpha,\gamma} c^{\alpha\gamma} \operatorname{Cov}_\beta \left[ \phi, \Phi_{\alpha\gamma} \right] .$$

*Proof.* We compute that

$$\Delta L = -\int (q'(y|x) - q(y|x))q(x) \log p(y|x, w) dx dy$$

$$= -\sum_{\alpha,\gamma} c^{\alpha\gamma} \int e_{\alpha\gamma}(x)(y)q(x) \log p(y|x, w) dx dy$$

$$= \sum_{\alpha,\gamma} c^{\alpha\gamma} \langle \Phi(w), e_{\alpha\gamma} \rangle_{\mathscr{H}}$$

so the conclusion follows from Lemma 2.2 and bilinearity of the covariance. $\qquad \square$

**Definition D.5.** For any $\alpha \in \Lambda, \gamma \in \Lambda^{++}$ we call

$$\chi_{\alpha\gamma} = -\operatorname{Cov}_\beta \left[ \phi, \Phi_{\alpha\gamma} \right]$$

the *susceptibility* of $\phi$ to the mode pair $\alpha, \gamma$.

This depends on the observable $\phi$ but not on the variation in the data distribution. From Lemma D.4 we obtain that the expression of the susceptibility in the mode basis is

$$\chi = \sum_{\alpha,\gamma} c^{\alpha\gamma} \chi_{\alpha\gamma} \qquad (29)$$

for some coefficients $c^{\alpha\gamma}$.

## D.5 STRUCTURAL INFERENCE AND MODES

In this section we provide motivation for the approach introduced in Section 3.3, by explaining how the theory of modes provides a factorization of the data matrix. We use the notation introduced in the main text and the previous section. We denote by $\chi_j^d$ the susceptibility for $q^d$ and observable $\phi_j$.

We denote by $\chi_{\alpha\gamma,j}$ the susceptibility for a mode pair $\alpha, \gamma$ and the observable $\phi_j$, as introduced in Appendix D.4. Ideally we would like to measure $\chi_{\alpha\gamma,j}$ but in practice we do not have the ability to sample from variations of the data distribution which only affect single modes. However, by Lemma D.4 (for mode-aligned concept shifts)

$$\chi_j^d = \sum_{\alpha \in \Lambda} c^{d,\alpha\gamma} \chi_{\alpha\gamma,j} \qquad (30)$$

where we note that $c^{d,\alpha\gamma}$ is from Definition D.2 and does not depend on $j$. This is a matrix equation

$$X = CP \qquad (31)$$

where $C$ is the $|\mathcal{D}| \times (|\Lambda||\Lambda^{++}|)$ matrix with entries $c^{d,\alpha\gamma}$ and $P$ is the $(|\Lambda||\Lambda^{++}|) \times |\mathcal{H}|$ matrix with entries $\chi_{\alpha\gamma,j}$. Then $C$ gives the coefficients of each variation of the data distribution in each mode and $P$ is the matrix of mode susceptibilities. More informally, we think of $C$ as the *coupling coefficients* between variations in the data and modes, and $P$ as the coupling coefficients between observables and modes.

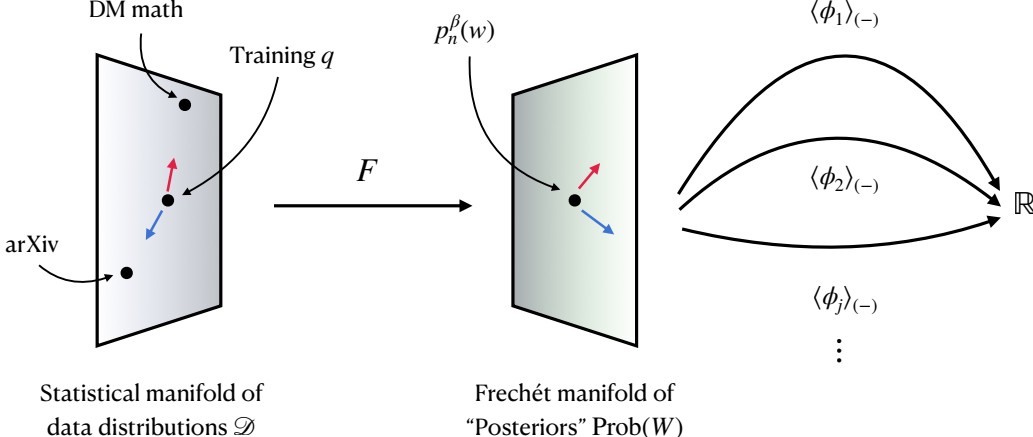

Figure 6: **Susceptibilities as components of a tangent map.** We can think of susceptibilities as components of the tangent map from data distributions to the expectations of a family of observables with respect to the posterior formed for that data distribution, with a fixed model. In this diagram we indicate $q \in \mathscr{D}$, the training distribution, and tangent vectors in this space corresponding to the mixtures with ARXIV and DM_MATHEMATICS. The tangent map of $F$ sends these tangent vectors to tangent vectors in the space of "posteriors".

The quantities $\chi_j^d$ may be estimated from data, that is, we can obtain an empirical estimate of the data matrix $X$. On the other hand $C, P$ are not directly observable because it is intractable to determine the complete set of modes of a complex distribution. However (31) suggests a way to apply methods of data analysis to the matrix $X$ in order to *infer* the modes and their couplings to the observables.

Applying matrix factorization methods like SVD to the data matrix $X$ is therefore one way to attempt to recover (linear combinations of) the underlying modes of the data distributions, or at least a coarse-graining of them. In this interpretation, the right singular vectors tell us about the relationship between attention heads and modes, and the principal components tell us about the coupling between modes and datasets.

### D.6    SUSCEPTIBILITIES AS A TANGENT MAP

This is all summarized neatly by Figure 6. We can define a smooth map $F$ which sends a distribution $q' \in \mathscr{D}$ over sequences of tokens to (for fixed $n, \beta$) the annealed and tempered posterior distribution $F(q') = p_n^\beta(w|q')$ as a point in the Frechét manifold $\mathrm{Prob}(W)$ of positive densities of integral 1 on $W$ (Bauer et al., 2016). Suitably well-behaved observables $\phi_j : W \longrightarrow \mathbb{R}$ define, by taking expectations, functions on this manifold. Given a set $\{\phi_j\}_{j \in \mathcal{H}}$ we therefore obtain a map

$$G = \begin{pmatrix} \langle \phi_1 \rangle_{(-)} \\ \vdots \\ \langle \phi_j \rangle_{(-)} \\ \vdots \end{pmatrix} : \mathrm{Prob}(W) \longrightarrow \mathbb{R}^{\mathcal{H}} .$$

Composing with $F$ gives a map $\mathscr{D} \longrightarrow \mathbb{R}^{\mathcal{H}}$ whose tangent map at $q$ is

$$T_q(G \circ F) = T_{p_n^\beta(w)}(G) \circ T_q(F) : T_q \mathscr{D} \longrightarrow T_{\langle \phi \rangle} \mathbb{R}^{\mathcal{H}} \tag{32}$$

where $\langle \phi \rangle$ denotes the vector of expectation values of all observables with respect to the unperturbed (annealed, tempered) posterior $p_n^\beta(w)$. By definition this linear map, or rather its restriction to the subspace of the tangent space spanned by the tangents to the mixtures with the $q^d$, has as its matrix the (transposed) response matrix $X^T$ (up to a factor of $n\beta$). The relationship between the matrix factorizations in (31) and (32) seems important.

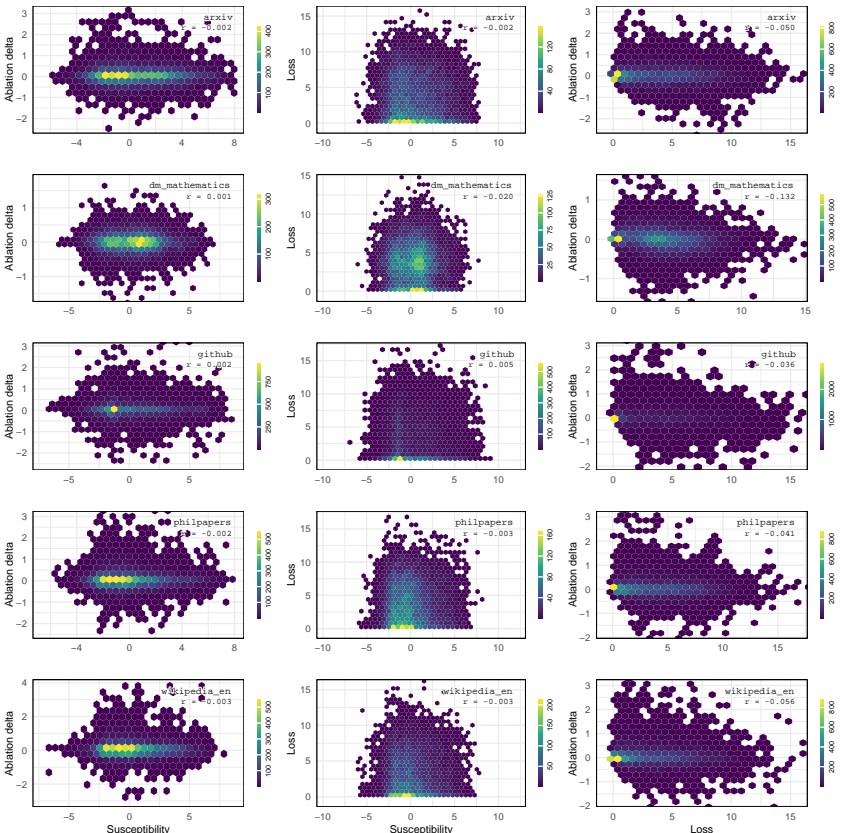

Figure 7: Pairwise comparisons of per-token susceptibilities, ablation, and loss as density maps for attention head 0:0. The color bar legend on the right of each subplot indicates the number of points represented by each colored hex.

# E SUPPORTING RESULTS

In this section we collect data and analysis that was performed to validate the basic methodology of susceptibilities, and which supports the headline results of the main text.

## E.1 PER-TOKEN METRIC COMPARISONS

In Figure 8 we select a pair of attention heads 0:0, 1:2 and a pair of datasets GITHUB, DM_MATHEMATICS and for data sampled from each dataset we compute the pair $(s, a)$ where $s$ is the per-token susceptibility and $a$ is the loss after ablation minus the pre-ablation loss (we call this the *ablation delta*). The correlation between these two metrics is very small.

In Figure 9, we see a specific example where the induction pattern is particularly visible in the positive susceptibilities, while there is no discernible induction pattern in the zero-ablation loss differences or original per-token loss.

## E.2 PER-TOKEN SUSCEPTIBILITY VARIATION WITH CONTEXT

In Figure 10 we plot the change in average per-token susceptibility for each attention head using 160 contexts from each of the following datasets: pile1m, arxiv, dm_mathematics, enron_emails, freelaw, github, nih_exporter, philpapers, pubmed_abstracts, pubmed_central, stackexchange, uspto_backgrounds, wikipedia_en.

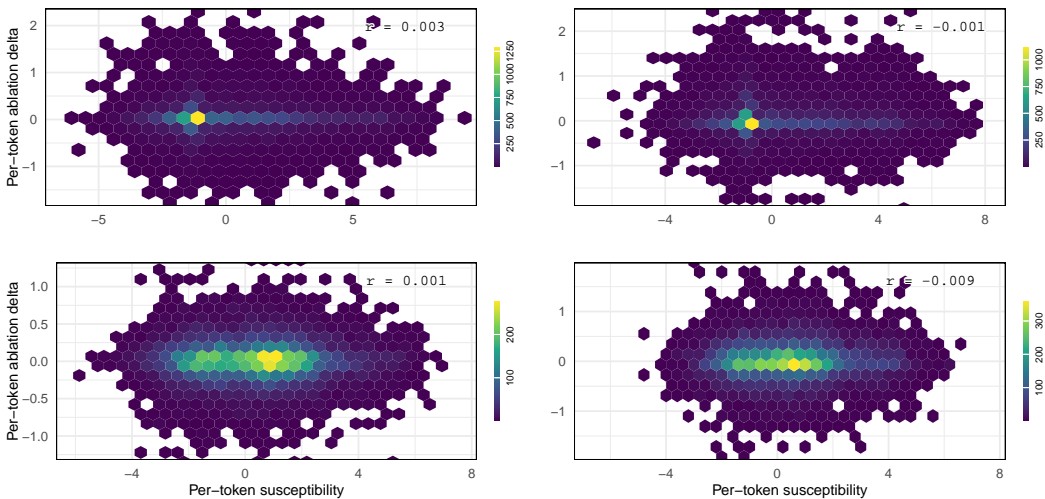

Figure 8: **Susceptibilities vs ablations.** Each plot shows distributions of per-token susceptibilities versus zero ablation loss differences as density maps. Each subplot corresponds to a pair of attention head and Pile subset, with datasets fixed across rows (top: GITHUB, bottom: DM_MATHEMATICS) and heads fixed across columns (left: 0:0, right: 1:2). Hexagon colors indicate a count of tokens with susceptibility and ablation delta values in a given region. Shown inset are the Pearson correlation coefficients, indicating negligible correlation.

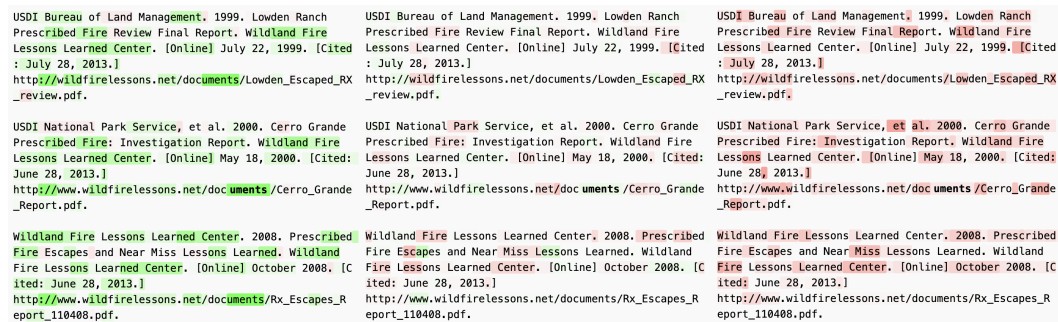

Figure 9: The same sample from freelaw is shown three times, from left to right with per-token susceptibilities, per-token ablation loss differences, and original token loss highlighted, with susceptibilities and ablations calculated for 1:4. On the left, green indicates a positive susceptibility and red a negative one. In the middle, green indicates a negative ablation loss difference (ablating the head improves performance) and red a positive ablation loss difference. On the right, a deeper red highlight indicates higher loss on that token.

We note that in general the per-token susceptibilities increase with context length, with the strongest rate of increase being in the layer 1 multigram heads 1:0-1:5 from length 10 onwards. It is not clear what the full explanation for this phenomenon is, but it is consistent with the observation that these heads tend to suppress induction patterns (Section 4.2) and the longer the context, the more token pairs $xy$ are recognizable as induction patterns.

### E.3    PER-TOKEN SUSCEPTIBILITIES EXPLAIN SUSCEPTIBILITY DIFFERENCES

It is natural to ask *why* some heads are outliers for particular datasets but not others. For instance 0:1 is a negative outlier (by which we mean that it lies below the other heads) for GITHUB but not FREELAW or HACKERNEWS (see Appendix F). Based on Appendix D.4 our model for susceptibilities is that a head is sensitive to some patterns in the data distribution more than others (we say it is

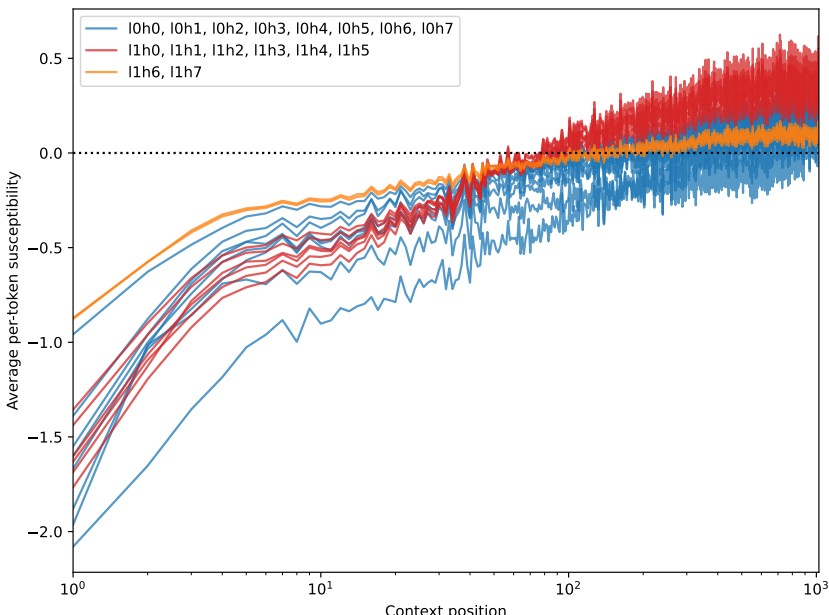

Figure 10: **Per-token susceptibilities as a function of context length.** The per-token susceptibilities for each attention head averaged over pairs $x, y$ where $x$ has a given length. Layer 0 heads are shown in blue, layer 1 induction heads in orange and the other layer 1 heads in red. We note a strong dependence on context length for these heads 1:0-1:5.

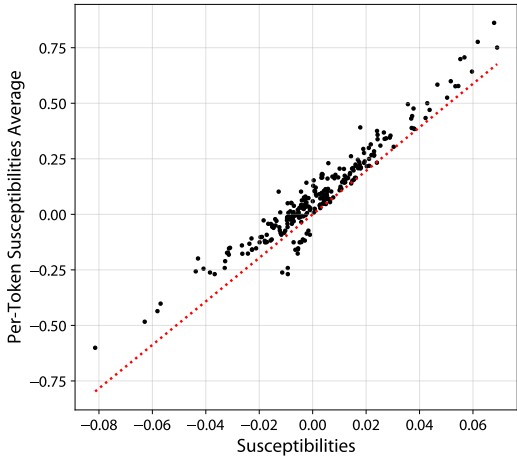

Figure 11: Comparison of average per-token susceptibilities and susceptibilities for every attention head and dataset. The dotted red line is the line of best fit.

*susceptible* to the pattern) and as we vary the data distribution this will drive different responses from different heads.

On this basis our hypothesis is that 0:1 is negatively susceptible to a pattern that is common in GITHUB but not in the other two datasets. To explore this we turn to the per-token susceptibilities (Definition 2.5) which we use to explain differences in overall susceptibilities. For this to be a sound methodology we need the averaged per-token susceptibilities to co-vary correctly with the susceptibility. In Figure 11 we plot one against the other. The Pearson's correlation coefficient is 0.958 and the line of best fit has slope 9.794. Since $\delta h = 0.1$ we would predict that the slope should be 10. We view this as sufficiently well-correlated that we can use per-token susceptibilities to study differences between susceptibilities.

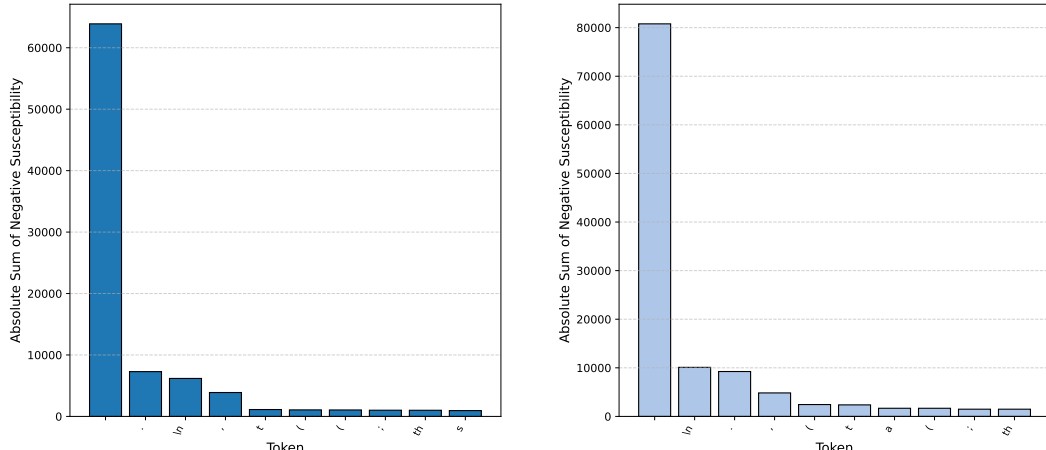

Figure 12: **Top negative susceptibility tokens.** Shown is the overall sum of negative susceptibility magnitude from 163k tokens sampled from GITHUB-ALL. *Left*: 0:0, *Right*: 0:1. In both cases the top contributing token is a space ▢.

Let us now turn to a detailed study of 0:0 versus 0:1 on GITHUB (at the end of training). For 163k tokens sampled from GITHUB-ALL we form the data in Table 2. Note that to obtain the scaled sum we add up the total positive and total negative susceptibility, compute the average over tokens by dividing by the total number of tokens ($163k$), and then multiply by $\delta h = 0.1$.

| Head | Total pos. | Total neg. | Sum | Scaled sum | ▢ Mean | ▢ Std |
|---|---|---|---|---|---|---|
| 0:0 | 171,561 | -145,342 | 26,219 | 0.016 | -1.49 | 0.52 |
| 0:1 | 139,458 | -205,255 | -65,797 | -0.040 | -1.89 | 0.73 |

Table 2: Total negative and positive per-token susceptibility for two attention heads on GITHUB-ALL and statistics for per-token susceptibilities $\chi_{(x,y)}$ with $y = $ ▢ the space token.

Comparing to the susceptibilities in Figure 28 we see that 0:0 lies in the range $[0.01, 0.02]$ and 0:1 is in the range $[-0.05, -0.06]$. The gap in total positive per-token susceptibility is 32k vs a gap of 60k in the negative susceptibility. Since we want to understand why 0:1 is a negative outlier, we focus on the latter gap. In Figure 12 we show the total contributions of individual tokens $y$ (over all contexts $x$) to the total negative susceptibility for the two heads. In both cases we see that the bulk comes from the space token ▢, which contributes roughly an order of magnitude more than the next most important token. In fact ▢ accounts for 44% of the total negative susceptibility for 0:0 and 39% for 0:1. The space token appears 42,764 times in this dataset, and the statistics of the per-token susceptibilities for ▢ (in any context) are shown in the rightmost two columsn of Table 2. The full distributions are shown in Figure 13.

The difference in the means is enough to explain about 17k of the total 60k gap in negative suscep-tibility, so about a third. This is evidence that a primary reason that 0:1 is a negative outlier on GITHUB-ALL is that this head is substantially more susceptible to the space token than other heads (e.g. it is 27% larger in magnitude than the mean susceptibility of 0:0) *and* spaces are more frequent in GITHUB than FREELAW or HACKERNEWS (compare 43k occurrences for GITHUB with 32k for FREELAW and 16k for HACKERNEWS). This is not surprising since code is often structured using space tokens.

### E.4 BIMODAL TOKENS

The per-token susceptibility $\chi_{(x,y)}$ is a function of a context $x$ and an individual token $y$ being predicted in that context. In Appendix E.3 we saw how differences in susceptibilities of heads can potentially be attributed to per-token susceptibilities, but there the analysis was of a token $y = $ ▢

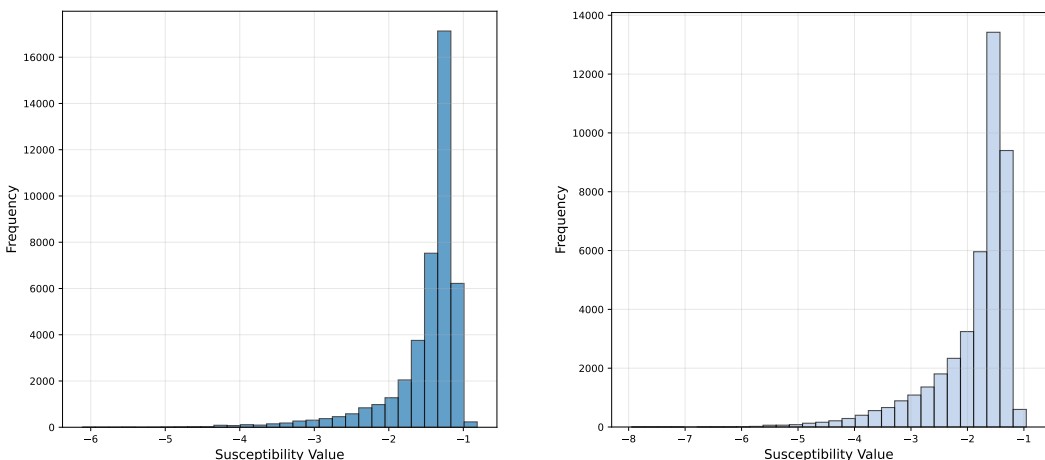

Figure 13: **Histogram of susceptibilities.** Shown is a histogram of per-token susceptibility values for the space token ☐ across 163k tokens sampled from GITHUB-ALL. *Left:* 0:0, *Right:* 0:1.

independent of its context $x$. Since not all occurrences of $y$ are semantically the same, we should expect that the distribution of susceptibilities across $x$ for a fixed $y$ should sometimes reflect this.

Indeed, in this section we report on an interesting phenomena where this distribution can be bimodal: there are some tokens $y$ for which there are two *kinds* of contexts $x$ with the susceptibilities $\chi_{(x,y)}$ in each context clustering around well-separated values.

**The head 0:0 and token `to`.** On both GITHUB-ALL and ARXIV we observe that the token `to` is bimodal for the attention head 0:0, as shown in Figure 14. On ARXIV there is one mode with positive susceptibility and another with negative susceptibility. A typical positive instance is

$$\text{app ears } \boxed{\text{to}}^{1.56} \text{ be relative ly ins ens itive}$$

where the overset number is the susceptibility of the outlined token. A typical instance of the negative mode is

$$\$ \text{ N } \backslash \boxed{\text{to}}^{-2.23} \backslash \text{ in ft y } \$ \text{ al ong .}$$

That is, the examples in the positive mode are normal occurrences of `to` as a word, whereas the negative mode consists of occurrences that are part of LaTeX commands. For GITHUB-ALL the token is also bimodal, with one positive mode and a negative mode. Typical instances of the positive mode are again occurrences of `to` in normal English sentences and a typical negative instance is

$$\text{if ( by tes \_ } \boxed{\text{to}}^{-2.17} \text{\_ s k ip = = - 1 ) .}$$

**The head 1:7 and token `/`.** Similarly we see in Figure 14 a positive and negative mode in the distribution of susceptibility values for 1:7 and the token `/` in HACKERNEWS. A typical instance of the positive mode is a forward slash in a URL:

$$\text{n pr . org } \boxed{/}^{4.10} \text{ se ctions } \boxed{/}^{2.35}$$

while typical negative instances are "either/or" constructions

$$\text{sa ving } \boxed{/}^{-2.76} \text{ com m itting , has h ic or p } \boxed{/}^{-3.61} \text{ go - m mult ier ror .}$$

Interestingly the distribution of susceptibilities is also bimodal for the same head and token on ENRON_EMAILS, with the positive mode still being about slashes in URLs, but the negative mode is closer to zero and occurs much more frequently (see Figure 15). Typical negative instances are

$$\text{K ay M ann } \boxed{/}^{-2.73} \text{ C or p } \boxed{/}^{-1.12} \text{ En ron , 11 } \boxed{/}^{-0.33} \text{ 09 } \boxed{/}^{1.60} \text{ 2 000 .}$$

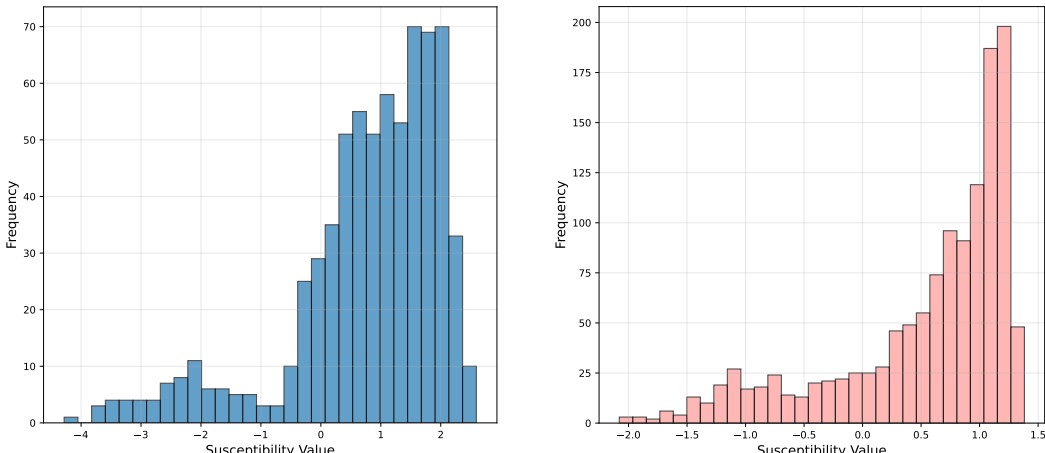

Figure 14: **Bimodal tokens.** Histograms showing frequency of certain susceptibility values for particular tokens and heads in particular datasets. *Left:* Attention head `0:0` and token `to` in ARXIV. *Right:* Attention head `1:7` and token `/` in HACKERNEWS.

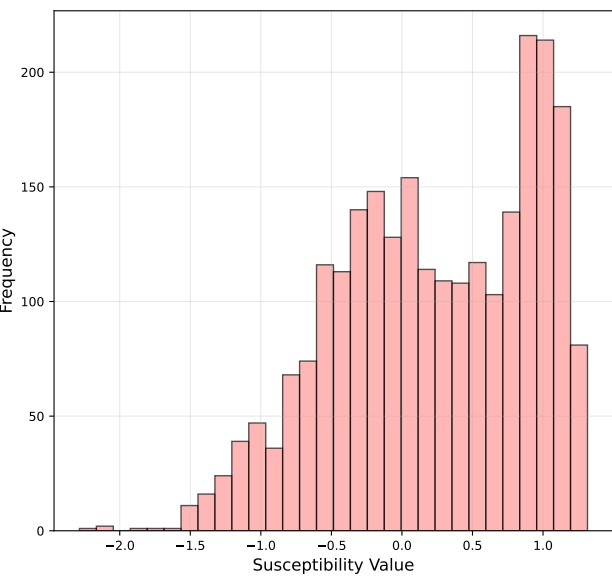

Figure 15: **Bimodal token.** Histograms showing frequency of certain susceptibility values for `1:7` and token `/` in ENRON_EMAILS.

### E.5 PER-TOKEN SUSCEPTIBILITY PCA

Here we collect some additional analysis of the PCs in Section 4.2.

In Figure 16 and Figure 17 we compare the complete distribution of susceptibilities for all heads on USPTO_BACKGROUNDS for all tokens vs just the induction pattern tokens, and we see here how the distribution for the layer 1 multigram heads (and to a significant but lesser extent 1:6) shifts to the right.

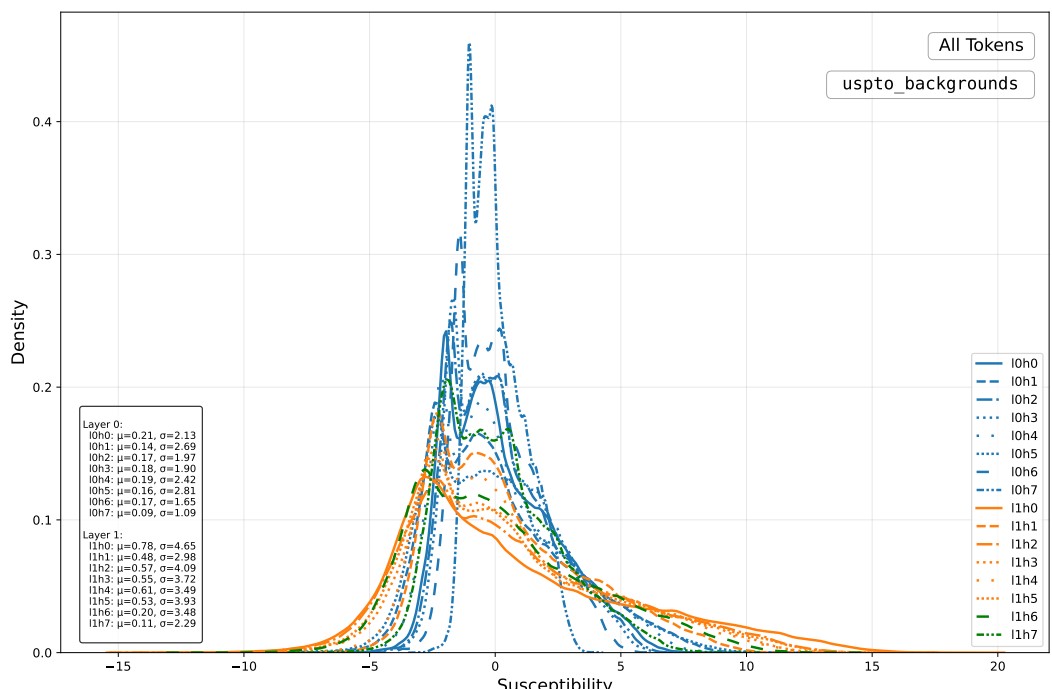

Figure 16: Distribution of per-token susceptibilities for all heads on USPTO_BACKGROUNDS, measured on all tokens. Curves are produced by KDE smoothing of data from 163k tokens.

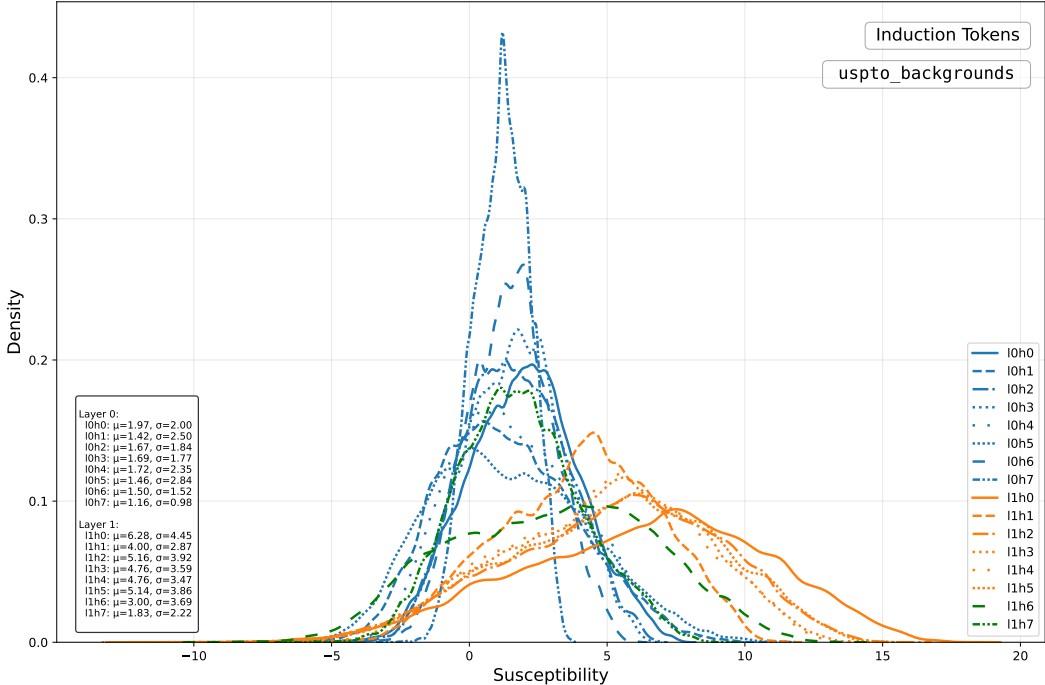

Figure 17: Distribution of per-token susceptibilities for all heads on USPTO_BACKGROUNDS, measured on just induction tokens. Curves are produced by KDE smoothing.

## F    TOP SUSCEPTIBILITIES

In this section we give additional details for each dataset (see Appendix B). The purpose of measuring susceptibilities is to use these "responses" to external perturbations in the data distribution to infer internal structure. Thus the first test for the methodology is whether or not the set of perturbations that we are using (e.g. Pile subsets) elicits a sufficient variety of responses. To see this it suffices to compare the plots of susceptibilities for each head across training on some datasets, for instance HACKERNEWS (Figure 27) and GITHUB-ALL (Figure 28).

Note that by definition the susceptibility with respect a variation of the Pile $q$ in the direction of dataset $q'$ is zero if $q'$ is distributed identically to $q$. Thus for subsets that are similarly distributed to the overall Pile we should expect susceptibilities to be small; see Appendix F.9.

Attention head $h$ in layer $l$ is denoted $\boxed{\text{l:h}}$. Tokens are denoted $\texttt{t}$ and the token with high (positive or negative) susceptibility is denoted with a border like $\boxed{\texttt{t}}$. In this section $x$ stands for a numeric token, e.g. $\texttt{4}$, $\texttt{13}$.

In each section we plot the susceptibility of each head for a particular dataset, and provide the three largest positive and negative susceptibility tokens taken from the one hundred tokens with largest susceptibility magnitude found in a sample of contexts. Note that these outlier tokens are not necessarily a good indicator of the "function" of a head.

The per-token susceptibility $\chi_{(x,y)}$ is a function of the predicted token $y$ and context $x$. We typically only give $y$ with at most some small segment of $x$, e.g.

$$\overbrace{\cdots x_{k-1} x_k}\quad \overset{y}{}$$
$$\boxed{\texttt{wa}}\ \boxed{\texttt{vel}}\ \boxed{\texttt{ength}}$$

where only the last two tokens of the context $x$ is given. In some cases we also show the largest magnitude negative per-token susceptibilities, with the same conventions. Recall from Section 3.2 that we interpret $\chi_{(x,y)} > 0$ for a particular head as a tendency for that head to act to *suppress* the prediction of $y$ in context $x$, so to a large extent this appendix is a "catalogue of suppression".

In the following $\texttt{://}$ always appears as part of $\texttt{http}\ \texttt{://}$ or $\texttt{https}\ \texttt{://}$ unless specified otherwise, and $\texttt{\#\#\#\#}$ appears as part of a "line" underneath a title or separating sections of a document. The token $\texttt{="}$ appears as part of HTML, e.g. $\texttt{class}\ \texttt{="}$ or similar markup.

### F.1    ARXIV

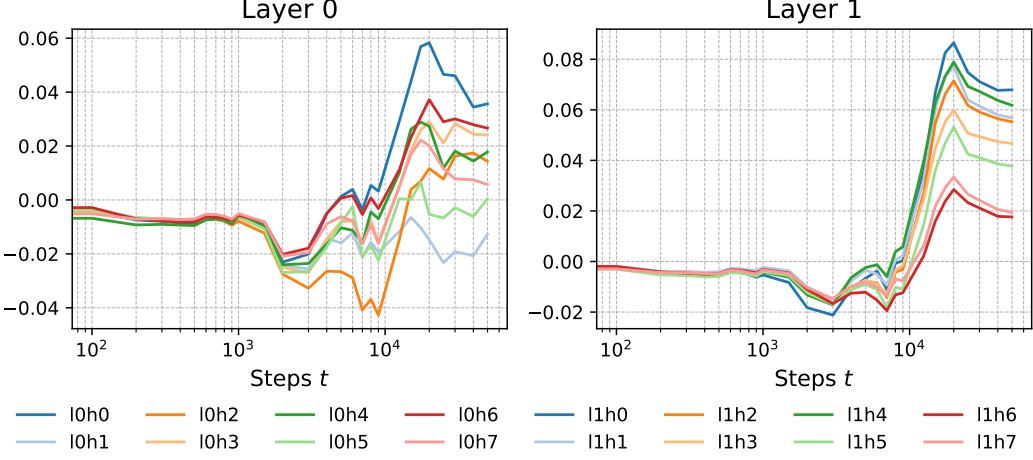

Figure 18: ARXIV susceptibilities for each head over training.

The susceptibilities for each head are shown in Figure 18.

Examples of positive per-token susceptibilities:

- `0:2`: ` wa vel `ength`, es `pecially`,  differe `nces`
- `1:4`: `="`,  ` wa vel `ength`, \ ome `ga`
- `1:7`: `="`, es `pecially`,  ` wa vel `ength`

Examples of negative per-token susceptibilities:

- `0:2`: ` \ `te` xt ,  $ `e` ^ ,  r `.`h . s`
- `1:4`: ` L `^` { ,`[` ** , \ `om` in us`
- `1:7`: `[` ** ,`<|endoftext|>`, \ [ `ree``

## F.2 DM_MATHEMATICS

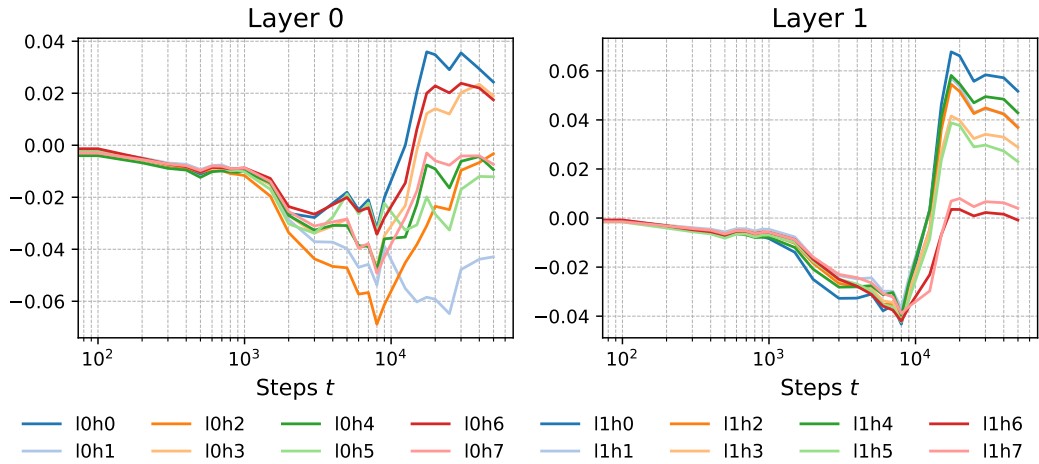

Figure 19: DM_MATHEMATICS susceptibilities for each head over training.

The susceptibilities for each head are shown in Figure 19.

Examples of positive per-token susceptibilities:

- `0:0`: `D eter `mine`, `}`, `)``
- `0:1`: `der iv `ative`,  repl ac `ement``
- `0:4`: `der iv `ative`, D eter `mine`,  repl ac `ement``

Examples of negative per-token susceptibilities:

- `0:0`: `[]`, ` is`, ` be``
- `0:1`: `[]`, ` is`, ` be``
- `0:4`: `[]`, ` is`, ` be``

## F.3 ENRON_EMAILS

The susceptibilities for each head are shown in Figure 20.

Examples of positive per-token susceptibilities:

- `0:3`: `="`, `://`, `####``
- `1:1`: `="`, `####`, `://``
- `1:7`: `="`, `://`,  ` el im `inate``

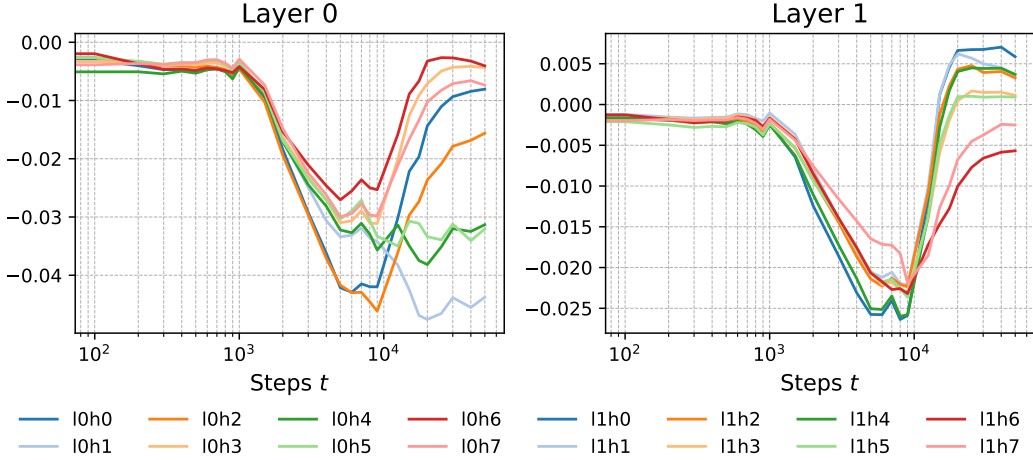

Figure 20: ENRON_EMAILS susceptibilities for each head over training.

Examples of negative per-token susceptibilities:

- 0:3: `<|endoftext|>`, `cc`, `.` `as` `p`
- 1:1: `<|endoftext|>`, `For`, `.` `as` `p`
- 1:7: `<|endoftext|>`, `[]`, `R` `ose`

## F.4  NIH_EXPORTER

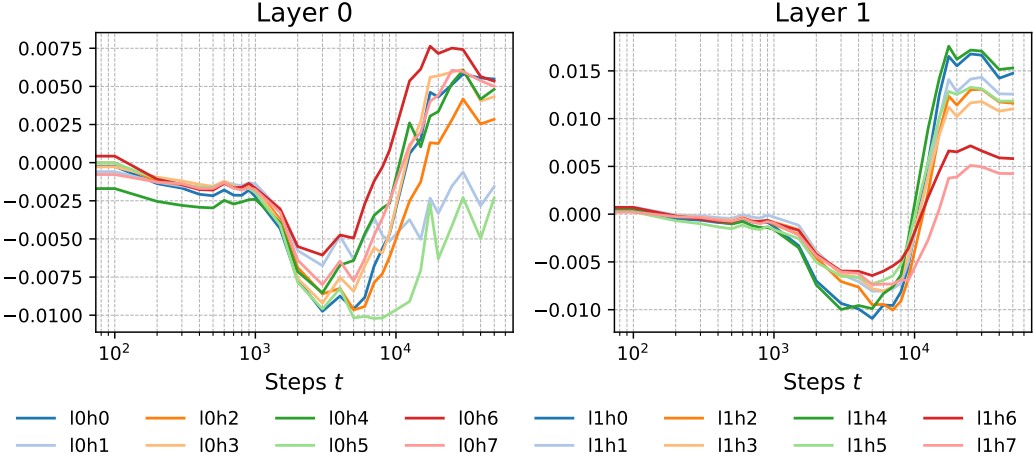

Figure 21: NIH_EXPORTER susceptibilities for each head over training.

The susceptibilities for each head are shown in Figure 21. Below $x$ stands for a token representing a number.

Examples of positive per-token susceptibilities:

- 0:6: ` differe` `nces`, ` ev` `al` `uate`, ` el` `im` `inate`
- 1:3: `th` `y` `roid`, `( x` `%)`, ` prot` `ot` `ype`
- 1:5: `th` `y` `roid`, ` el` `im` `inate`, ` full` `− l` `ength`

Examples of negative per-token susceptibilities:

- 0:6: qual i ty , ; , obvious
- 1:3: qual i ty , ; , (
- 1:5: qual i ty , ; , Ca 2 ion

### F.5 PUBMED_ABSTRACTS

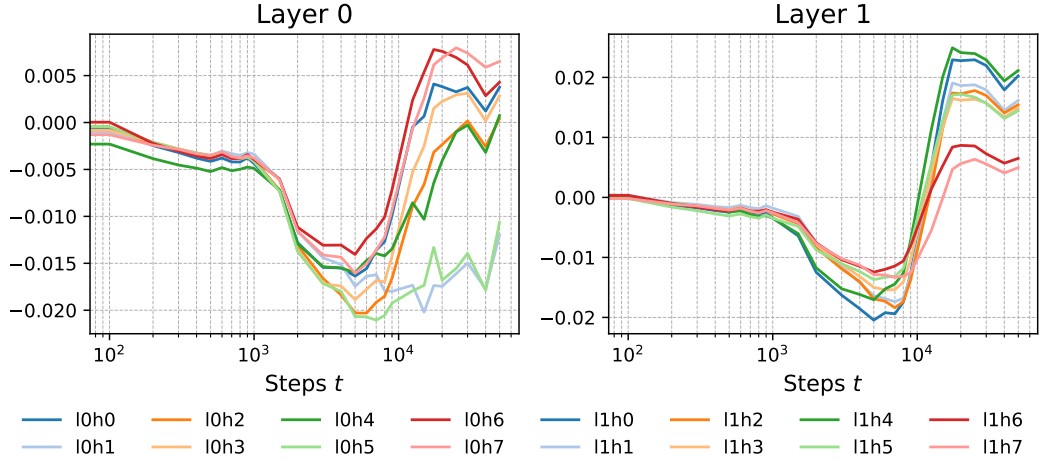

Figure 22: PUBMED_ABSTRACTS susceptibilities for each head over training.

The susceptibilities for each head are shown in Figure 22. Below $x$ stands for a token representing a number.

Examples of positive per-token susceptibilities:

- 0:0: C a ++ , ( x % ), differe nces
- 0:5: ( x % ), demon st ration , differe nces
- 1:5: C a ++ , Th y roid , ( x % )

Examples of negative per-token susceptibilities:

- 0:0: med ium , , pub oper ine al is
- 0:5: ili oc oc cy ge ous , ol id ined ion es ,
- 1:5: , sc ro ful ace um , ev al u ations

### F.6 PUBMED_CENTRAL

The susceptibilities for each head are shown in Figure 23.

Examples of positive per-token susceptibilities:

- 0:1: :// , differe nces , ====
- 1:4: full – l ength , =" , ://
- 1:7: :// , =" , full – l ength

Examples of negative per-token susceptibilities:

- 0:1: pro ins ul in , , end ere r nd r one
- 1:4: k in ase , ect od er m al , l un ul ate
- 1:7: pro ins ul in , k in ase ,

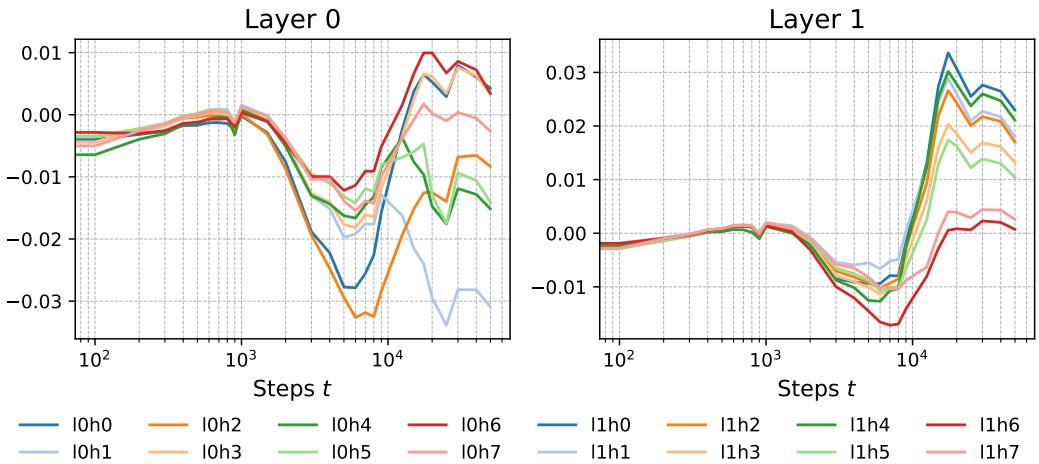

Figure 23: PUBMED_CENTRAL susceptibilities for each head over training.

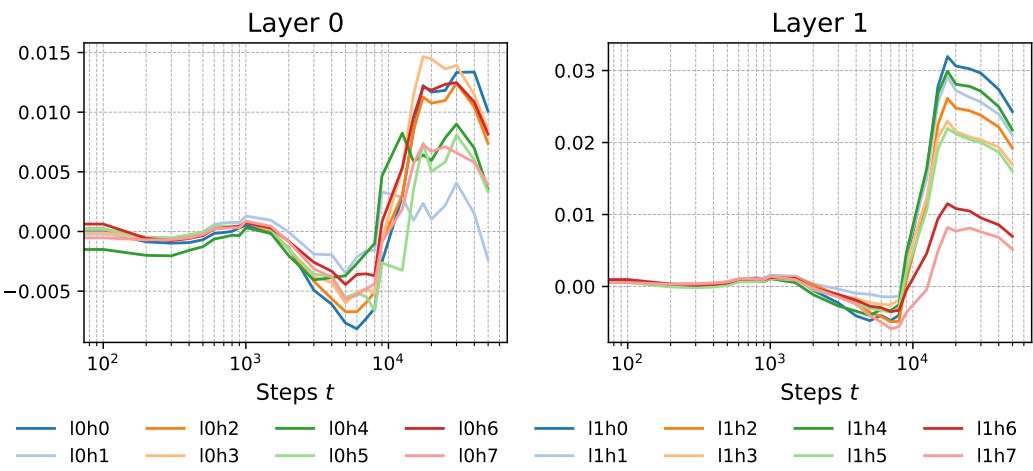

Figure 24: USPTO_BACKGROUNDS susceptibilities for each head over training.

### F.7 USPTO_BACKGROUNDS

The susceptibilities for each head are shown in Figure 24.

Examples of positive per-token susceptibilities:

- `0:2`: `wa` `vel` `ength`, `es` `pecially`, `differe` `nces`
- `1:2`: `wa` `vel` `ength`, `S` `od` `er` `berg`, `mag` `n` `itude`
- `1:4`: `wa` `vel` `ength`, `mag` `n` `itude`, `.` `ht` `ml`

Examples of negative per-token susceptibilities:

- `0:2`: `()`, `of`, `color`
- `1:2`: `()`, `color`, `of`
- `1:4`: `met` `er`, `color`, `()`

### F.8 WIKIPEDIA_EN

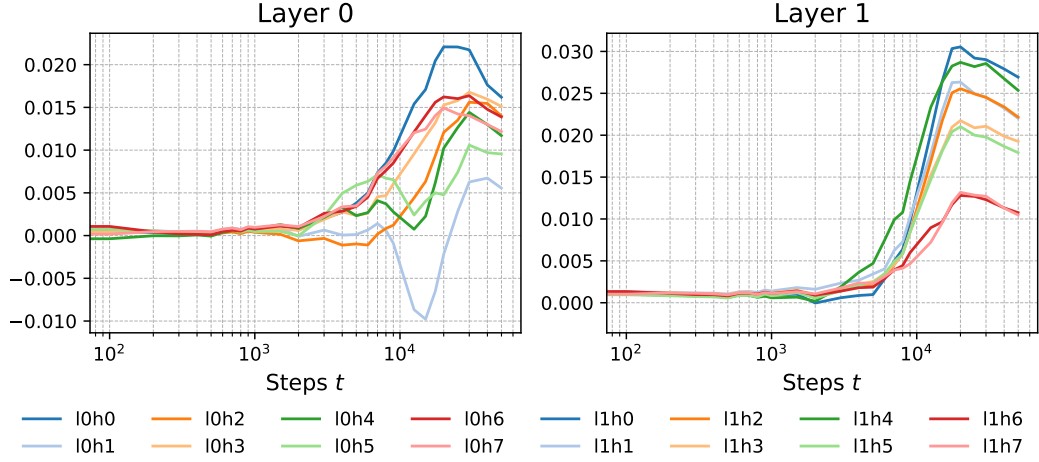

Figure 25: WIKIPEDIA_EN susceptibilities for each head over training.

The susceptibilities for each head are shown in Figure 25.

Examples of positive per-token susceptibilities:

- `0:0`: `://`, `R` `ef` `erences`, `differe` `nces`
- `1:3`: `R` `ef` `erences`, `al` `bum`, `Y` `is` `rael`
- `1:4`: `E` `k` `berg`, `://`, `al` `bum`

Examples of negative susceptibilities:

- `0:0`: `in`, `<|endoftext|>`, `on`
- `1:3`: `<|endoftext|>`, `record`, `on`
- `1:4`: `<|endoftext|>`, `of`, `record`

### F.9 CC

The susceptibilities for each head are shown in Figure 26. Instances of `":"` typically occur as part of dictionaries, e.g. `{"` `name` `":"` but the preceding tokens vary.

Examples of positive per-token susceptibilities:

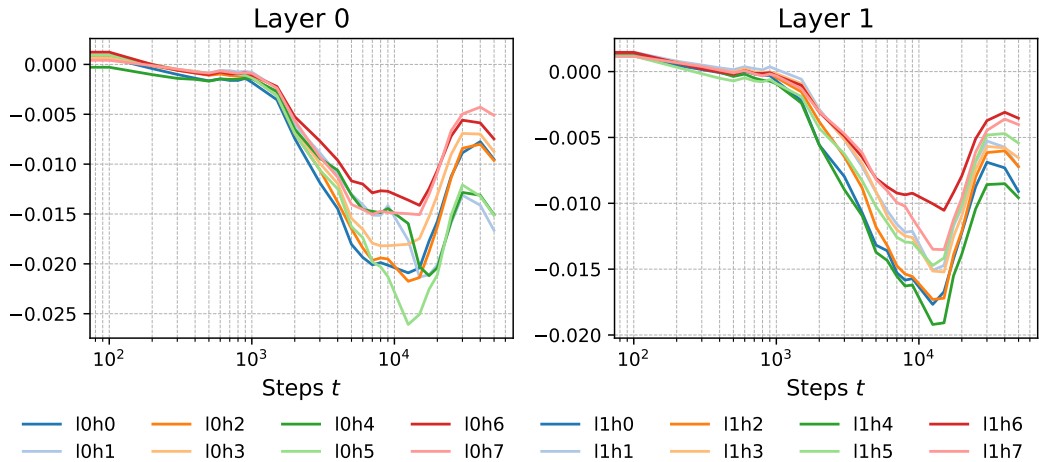

Figure 26: PILE-CC susceptibilities for each head over training.

- [0:1]: [th] is , [ more ], [ from ]
- [0:7]: [23], [and], []
- [1:6]: g all [ery], O ly mp [ic], under [ground]

Examples of negative per-token susceptibilities:

- [0:1]: f [il] m f est ival . com , [a], []
- [0:7]: f [il] m f est ival . com , [a], st [one]
- [1:6]: [a], [ games ], st [one]

We note that the susceptibilities for heads on CC are small relative to other datasets. Note that by definition the susceptibility should be zero for a variation $q \to q'$ where $q'$ is distributed identically to $q$, so this could be because CC is a significant fraction of the Pile (18% according to Gao et al. (2020)). In Table 3 we give a variant of Table 2 where we compute the total positive and negative per-token susceptibility of [0:0] on the datasets CC, GITHUB-ALL.

| Dataset | Total pos. | Total neg. | Sum | Scaled sum |
|---|---|---|---|---|
| GITHUB-ALL | 171,561 | -145,342 | 17,816 | 0.016 |
| CC | 128,026 | -130,095 | -966 | -0.0013 |

Table 3: Total negative and positive per-token susceptibility for [0:0] on two datasets, and the sum divided by the total number of tokens and multipled by $\delta h = 0.1$.

We note that the total positive and negative per-token susceptibilities are of a similar order of magnitude for both datasets, but their difference is an order of magnitude smaller (966 vs 17816) and this explains the order of magnitude difference in the final susceptibility. Recall that macroscopic bodies have near zero electric charge because of the exact cancellation of very *large* amounts of positive and negative charge in physical matter at equilibrium; it seems a reasonable intuition that near zero susceptibilities are likewise caused by near-exact cancellations rather than both positive and negative per-token susceptibilities being small.

Also note that, given the similarity between the tokens with top positive susceptibility shown above, it is no surprise that Figure 26 shows little differentiation of the heads. This particular variation in the data distribution away from the full Pile is not enough to "break the symmetry" between the heads, by exposing them to distributions of patterns for which they have distinct preferences.

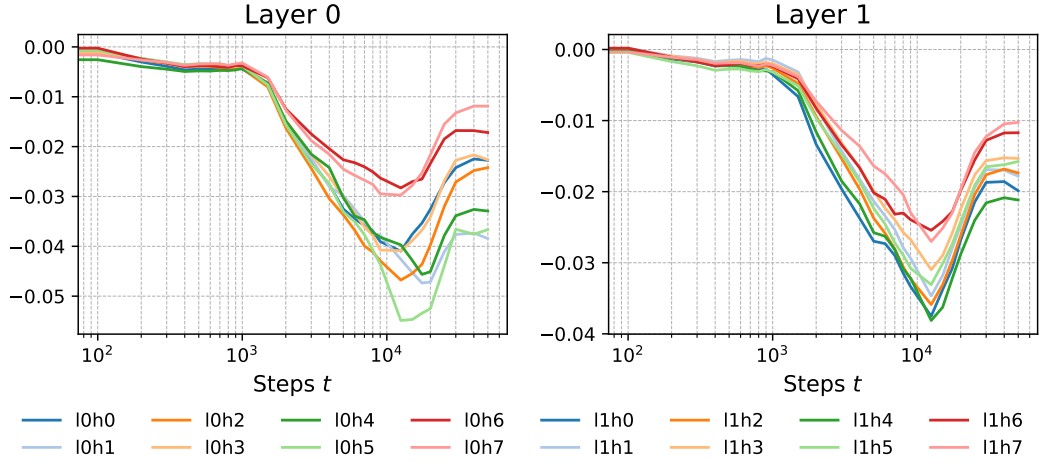

Figure 27: HACKERNEWS susceptibilities for each head over training.

## F.10 HACKERNEWS

The susceptibilities for each head are shown in Figure 27.

Examples of positive per-token susceptibilities:

- 0:0 : I ['ve], [ I ], [ just ]
- 1:1 : [], al [one], I ['ve]
- 1:5 : [], [ I ], I ['ve]

Examples of negative per-token susceptibilities:

- 0:0 : [ recommend ], [and], [ mon ] et ary
- 1:1 : [and], [the], [ co ] ff ee
- 1:5 : [and], [ recommend ], [ mon ] et ary

## F.11 GITHUB-ALL

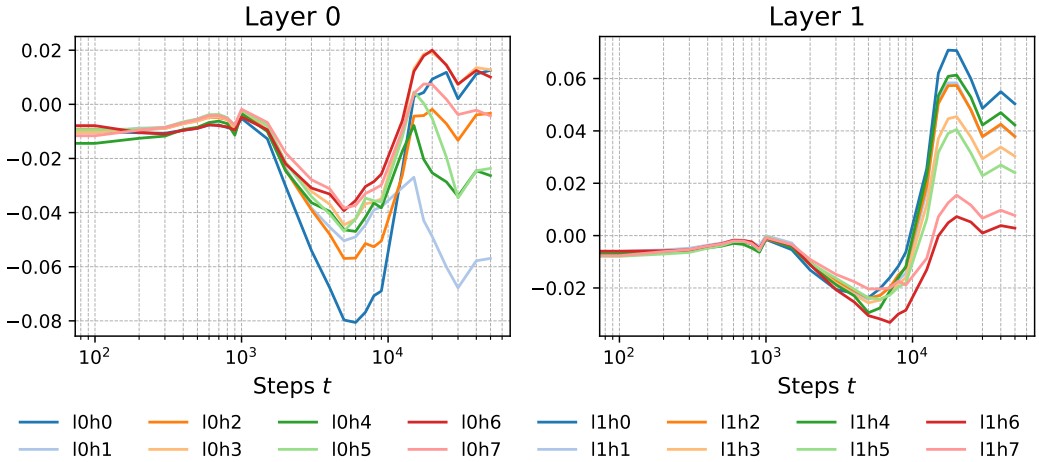

Figure 28: GITHUB-ALL susceptibilities for each head over training.

The susceptibilities for each head are shown in Figure 28.

Examples of positive per-token susceptibilities:

- 0:0 : =" , Get St ring L ength , ()
- 0:1 : :// , =" , count
- 1:6 : =" , :// , . l ength

Examples of negative per-token susceptibilities:

- 0:0 : () , <|endoftext|> , int count
- 0:1 : def connect _ , int count , con st base
- 1:6 : con st base , con st st d , int count

## F.12   FREELAW

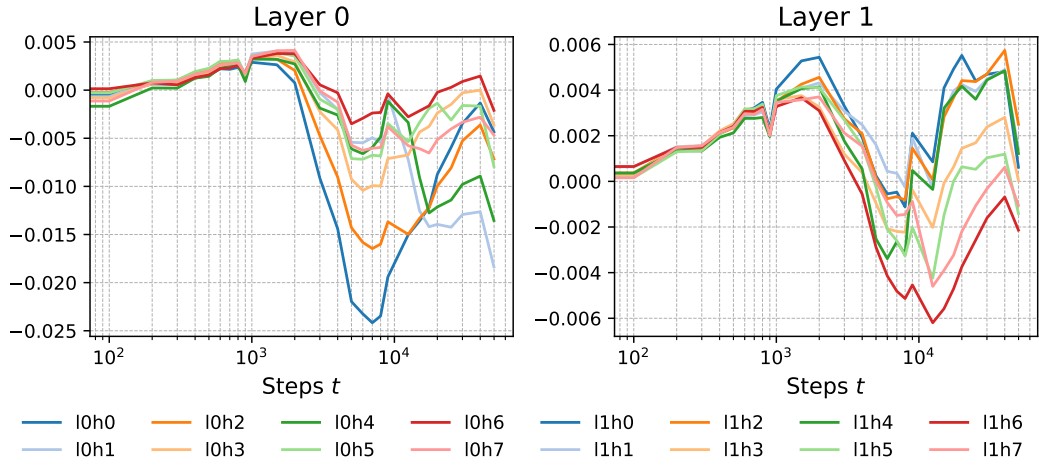

Figure 29: FREELAW susceptibilities for each head over training.

The susceptibilities for each head are shown in Figure 29.

Examples of positive per-token susceptibilities:

- 0:6 : :// , An al ysis , differe nces
- 1:4 : :// , O d ys sey , d oc uments
- 1:5 : :// , O d ys sey , K ap n icks

Examples of negative per-token susceptibilities:

- 0:6 : on , rem − \n edy , defense
- 1:4 : C . H , A ri − \n z ona , S up . Ct
- 1:5 : Rep resent a − t ives , C . H , S up . Ct

We note here the role of the − token as splitting a word across multiple lines, and . in abbreviations. In the negative susceptibilities there are several other examples of the tokens being used in other split words and abbreviations respectively.

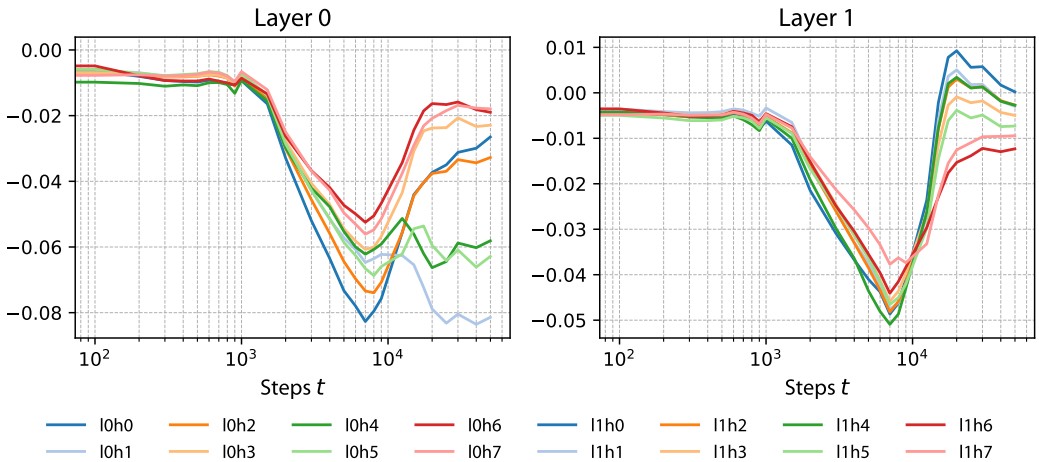

Figure 30: STACKEXCHANGE susceptibilities for each head over training.

### F.13 STACKEXCHANGE

The susceptibilities for each head are shown in Figure 30.

Examples of positive per-token susceptibilities:

- `0:2`: `://`, `="`, D ou ble `>`
- `0:4`: `://`, `="`,  differe `nces`
- `1:3`: `="`, `":"`, `()`

Examples of negative per-token susceptibilities:

- `0:2`: ` with`,  H T M LE `lement`, `[]`
- `0:4`:  H T M LE `lement`, ` with`,  b `in`
- `1:3`: `[]`,  H T M LE `lement`, ` where`

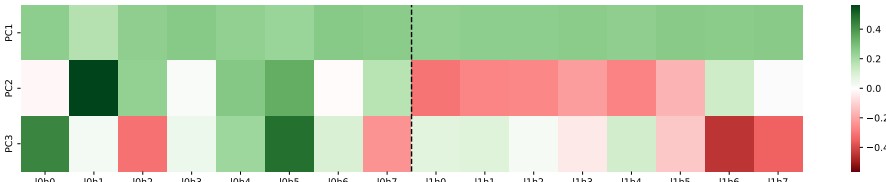

Figure 31: **Loadings of PCs on heads**. The loadings of each principal component on each head is shown, with the first principal component being a uniform mode, the second principal component loading (in layer 1) on the multigram heads, and the third principal component loading on the induction heads and 1:5.

## G  STRUCTURAL INFERENCE VIA TRAJECTORY PCA

In the main text (Section 4.2) we do structural inference on our small language model using PCA on a data matrix of per-token susceptibilities at the end of training. To provide an independent check on the basic results we provide in this appendix details of an alternative analysis using PCA of a data matrix of susceptibilities (not per-token) of heads *over* training. First we explain the methodology (Appendix G.1) and then present the results (Appendix G.2).

### G.1  JOINT TRAJECTORY PCA

We adapt a multi-trajectory variant of trajectory PCA (Amadei et al., 1993; Briggman et al., 2005) to study how the behaviors of attention heads in our small language model evolve with respect to different data distributions; see also Carroll et al. (2025, §4.2).

For each attention head $j \in \mathcal{H}$ and each dataset $d \in \mathcal{D}$, we measure the susceptibility $\chi_j^d(t)$ at each training checkpoint $t \in \mathcal{T}$. The checkpoints used are the same ones given in Appendix C.3. This gives us, for each dataset $d$, a trajectory through a space with dimensions corresponding to attention heads $\gamma_d(t) = \left(\chi_1^d(t), \chi_2^d(t), \ldots, \chi_{|\mathcal{H}|}^d(t)\right) \in \mathbb{R}^{|\mathcal{H}|}$. We combine these trajectories into a data matrix $X \in \mathbb{R}^{|\mathcal{D}||\mathcal{T}| \times |\mathcal{H}|}$ where each row represents the susceptibility measurements across all attention heads for a specific dataset at a specific checkpoint. For each $d \in \mathcal{D}$, we aggregate the row vectors from each checkpoint into a matrix $X_d \in \mathbb{R}^{|\mathcal{T}| \times |\mathcal{H}|}$ and then stack each $X_d$ vertically into $X \in \mathbb{R}^{|\mathcal{D}||\mathcal{T}| \times |\mathcal{H}|}$:

$$X_d = \begin{bmatrix} \gamma_d(t_1) \\ \gamma_d(t_2) \\ \vdots \\ \gamma_d(t_{|\mathcal{T}|}) \end{bmatrix} \text{ for } d \in \mathcal{D}, \quad X = \begin{bmatrix} X_{d_1} \\ \vdots \\ X_{d_{|\mathcal{D}|}} \end{bmatrix}.$$

We standardize each column of $X$ to have zero mean and unit variance, ensuring that attention heads with larger susceptibility magnitudes do not dominate the analysis. Let $X_{\text{std}}$ denote this standardized matrix. We then perform singular value decomposition (SVD)

$$X_{\text{std}} = U\Sigma V^T \tag{33}$$

where $U \in \mathbb{R}^{|\mathcal{D}||\mathcal{T}| \times c}$ contains the left singular vectors, $\Sigma \in \mathbb{R}^{c \times c}$ is a diagonal matrix of singular values, and $V \in \mathbb{R}^{c \times |\mathcal{H}|}$ contains the right singular vectors where $c$ is the chosen number of principal components. The principal components are given by the columns of $U\Sigma$, and the loadings that indicate how attention heads contribute to each principal component are given by the rows of $V^T$.

For each dataset $d$, we project its trajectory $\gamma_d(t)$ onto the reduced space defined by the first $k$ principal components $\pi_d(t) = (\gamma_d(t) - \mu) \cdot V_k$ where $\mu$ is the mean vector used during standardization and $V_k$ consists of the first $k$ columns of $V$. This gives us a low-dimensional representation $\pi_d(t) \in \mathbb{R}^k$ of the trajectory for dataset $d$ at checkpoint $t$.

### G.2  RESULTS

We perform joint trajectory PCA as a concrete realisation of the structural inference proposed in Section 3.3. We found that three principal components explained 98% of the variance (with the top

three PCs explaining respectively $85.78\%, 10.76\%, 1.63\%$). In more detail, the loadings of the PCs as presented in Figure 31 are:

- **PC1** is a uniform mode, loading on all heads.
- **PC2** loads strongly positively on `0:1` and negatively on `1:0` - `1:5`, that is, the layer 1 multigram heads.
- **PC3** loads strongly positively on `0:0`, `0:5` and negatively on `0:2`,`0:7` in layer 0 and strongly negatively on `1:6`, `1:7` in layer 1.

We see therefore that PC2 separates the layer 1 multigram heads and induction heads, as noted in Section 4.2 for the per-token susceptibility PC2. Since the susceptibility for a head is the expectation over the probe data distribution of the per-token susceptibilities, it is not a surprise to see that there is broad agreement between the structure revealed by the trajectory PCA and the analysis in Section 4.2. Nonetheless the switch from per-token to overall susceptibility, and the inclusion of multiple checkpoints, provides a non-trivial check on the robustness of that analysis.

## H COMPARISON TO DIRECT LOGIT EFFECTS

Previous work (Gurnee et al., 2024b) (Lad et al., 2025) has defined prediction and suppression neurons with reference to $W_U W_O^h$, where $W_O^h$ is the output matrix for head $h$, and $W_U$ the unembedding matrix for the network, following Elhage et al. (2021). Entry $(i, j)$ is interpreted as the direct contribution of neuron $j$ in head $h$ to token $i$ in our vocabulary $\Sigma$. In particular, one says a neuron at index $j$ in head $h$ is predictive when the effect distribution $(W_U W_O^h)_{:,j}$ has a high kurtosis and positive skew, and suppressive when the effect distribution has a high kurtosis and negative skew.

Given our own characterization of negative susceptibilities as indicating expression, one hypothesis is that a negative susceptibility for head $h$ on a token $t$ means that neurons in head $h$ typically have a positive effect on the logits of $t$, while a positive susceptibility indicates neurons typically have a negative effect.

Let $X = \{(x_1, y_1), \ldots, (x_n, y_n)\} \in D_N$ be $n$ samples for which we have computed $\chi_{(x_i, y_i)}$ from our dataset $D_N$, and $U = \{u_1, \ldots, u_m\} \subseteq \Sigma$ be the set of unique $y_i$'s in this subset. Then for each $u \in U$ define $X_{y=u} = \{(x_j, y_j) \in X | y_j = u\} = \{(x_{j_1}, y_{j_1}), \ldots, (x_{j_k}, y_{j_k})\}$, and $\overline{\chi}_{y=u}^h = \frac{1}{k} \sum_{i=1}^k \chi_{(x_{j_i}, y_{j_i})}^h$, where $\chi_{(x,y)}^h$ is the per-sample susceptibility of head $h$ to sample $(x, y)$, so that $\overline{\chi}_{y=u}^h$ is the average per-sample susceptibility on samples with the completion $u$.

For each attention head $h$ and every token $u \in U$ we plot the quantity $\overline{\chi}_{y=u}^h$ against the average value of $(W_U W_O^h)_{u,:}$. The results for each head are shown in Figure 32 along with the corresponding $r^2$ and slope of each graph.

We note that both the $r^2$ values and the slopes are very small, with slopes having inconsistent signs, rejecting the hypothesized connection.

## I ADDITIONAL SEEDS

Three additional models were trained with the same architecture and training data, but different random seeds; we refer to these as seeds $2, 3, 4$ (with the original being seed 1).

## J VARIANCE OF PRINCIPAL COMPONENTS

In Figure 2 and Figure 3 of the main text we present the data and component loadings of the top principal components of the per-token susceptibility PCA. Recall that this is done separately for the two layers. A prerequisite for assigning meaning to these principal components is that they are suitably stable with respect to the tokens chosen.

In this appendix we present the top five principal components of the per-token susceptibility PCA for both layers, including data for all nine patterns identified in Appendix C.8 as well as information

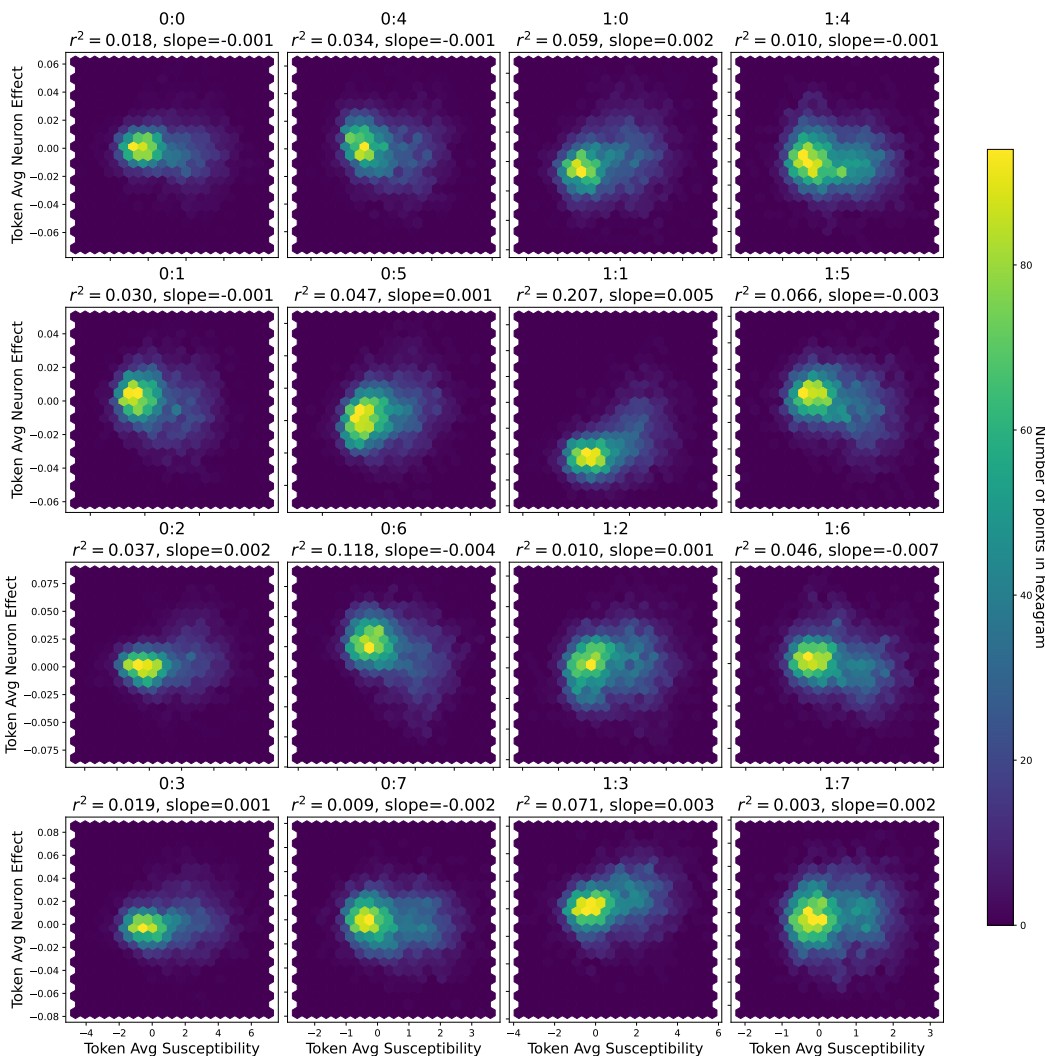

Figure 32: **Average token susceptibility plotted against average neuron effect.** Hex bin plots for each head $l : h$ plotting the points $(\overline{\chi}_{y=u_1}^{l:h}, \overline{(W_U W_O^{l:h})_{u_1,:}}), \ldots, (\overline{\chi}_{y=u_m}^{l:h}, \overline{(W_U W_O^{l:h})_{u_m,:}})$ for each $u_i \in U$ and with $\overline{(W_U W_O^{l:h})_{u,:}}$ indicating the average value of $(W_U W_O^{l:h})_{u,:}$. The head, $r^2$ value, and slope are indicated above each plot, and hexagrams are coloured according to the number of such points which lie inside them.

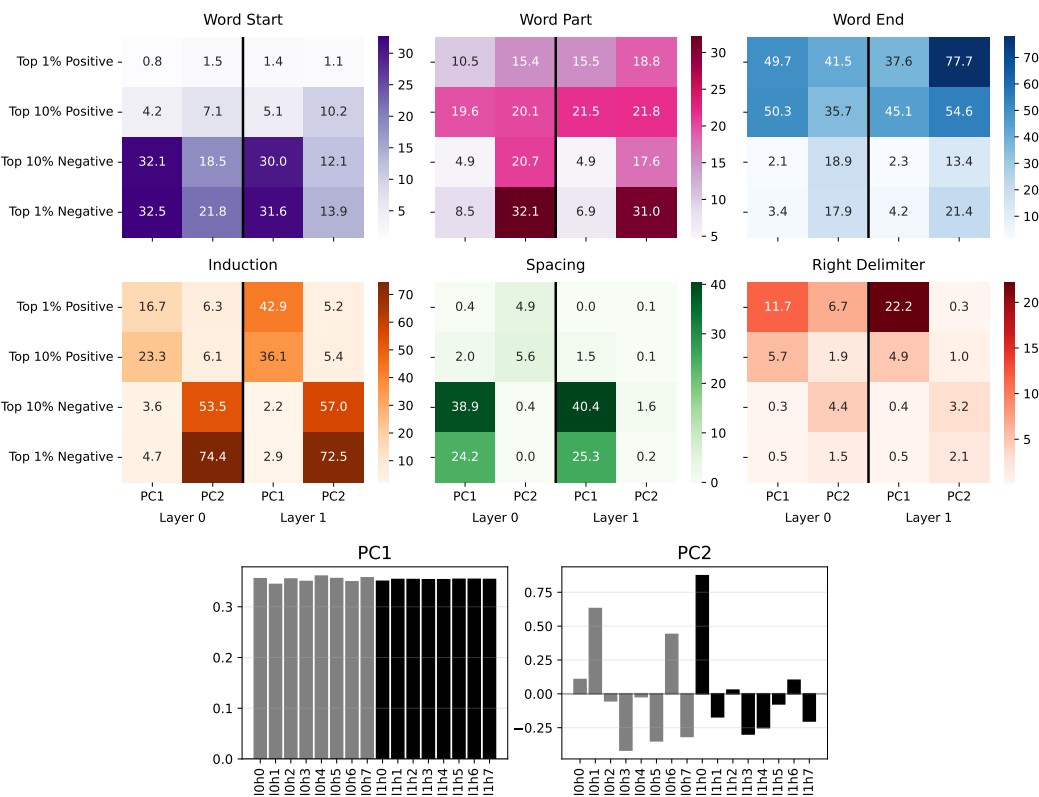

Figure 33: **Per-token susceptibility PCAs for Seed** 2. Among the tokens with the coefficients of the largest magnitude in each principal component for the per-token susceptibility PCA, the percentage following each of the six patterns (*Top*). The loadings of the principal components on heads (*Bottom*). In Wang et al. (2024, Appendix G) it was found that in this seed the previous-token head is 0:1, the current-token head is 0:6 and the induction head is 1:0.

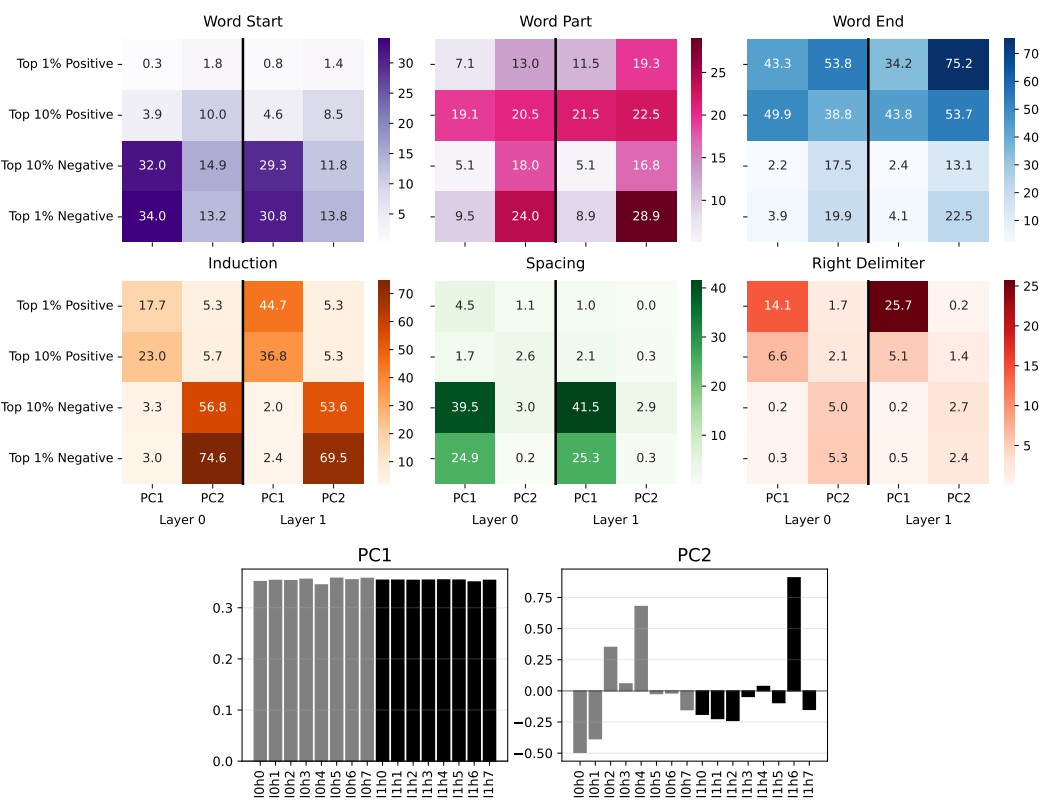

Figure 34: **Per-token susceptibility PCAs for Seed** 3. Among the tokens with the coefficients of the largest magnitude in each principal component for the per-token susceptibility PCA, the percentage following each of the six patterns (*Top*). The loadings of the principal components on heads (*Bottom*). In Wang et al. (2024, Appendix G) it was found that in this seed the previous-token head is 0:4, the current-token head is 0:2 and the induction head is 1:6.

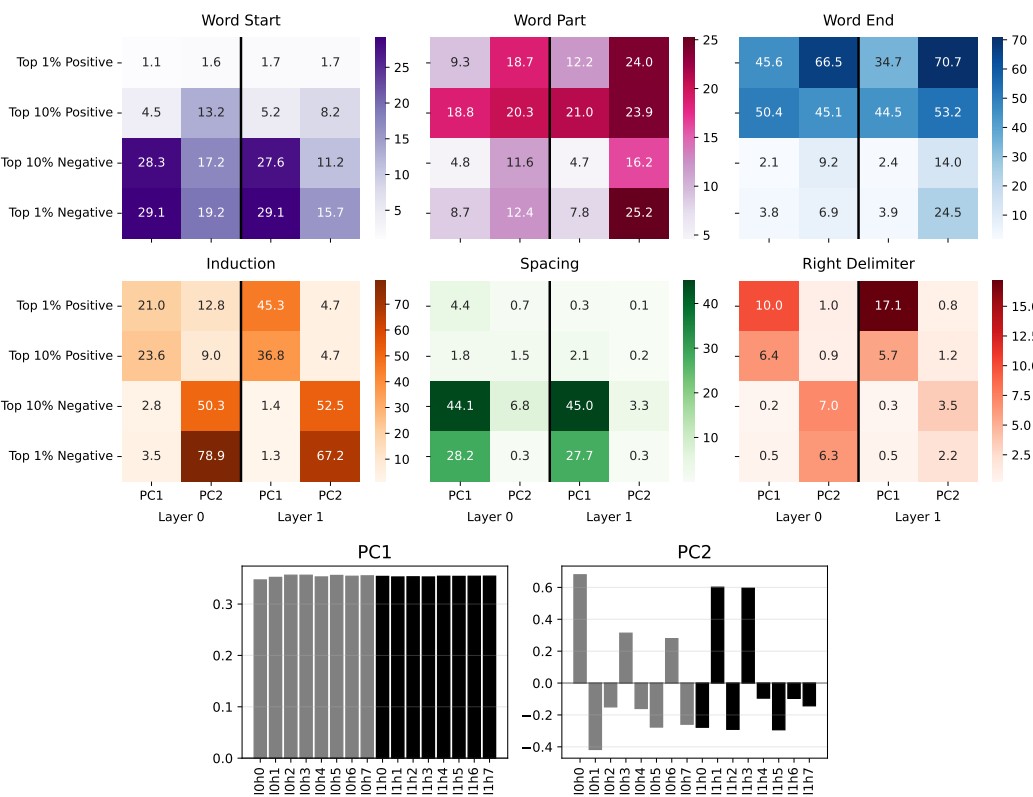

Figure 35: **Per-token susceptibility PCAs for Seed** 4. Among the tokens with the coefficients of the largest magnitude in each principal component for the per-token susceptibility PCA, the percentage following each of the six patterns (*Top*). The loadings of the principal components on heads (*Bottom*). In Wang et al. (2024, Appendix G) it was found that in this seed the previous-token heads are 0:0 and 0:3, the current-token head is 0:6 and the induction heads are 1:1 and 1:3.

about sample variance. As in the rest of the paper (with the exception of Appendix I) all data is for seed 1. The stochasticity here is the 20000 sampled tokens from each dataset. We report the mean and standard deviation of the percentages and head loadings in Figure 36, Figure 37 for layer 0 and Figure 38, Figure 39 for layer 1.

The conclusion is that the sample variance is sufficiently small to justify the interpretations given in the main text.

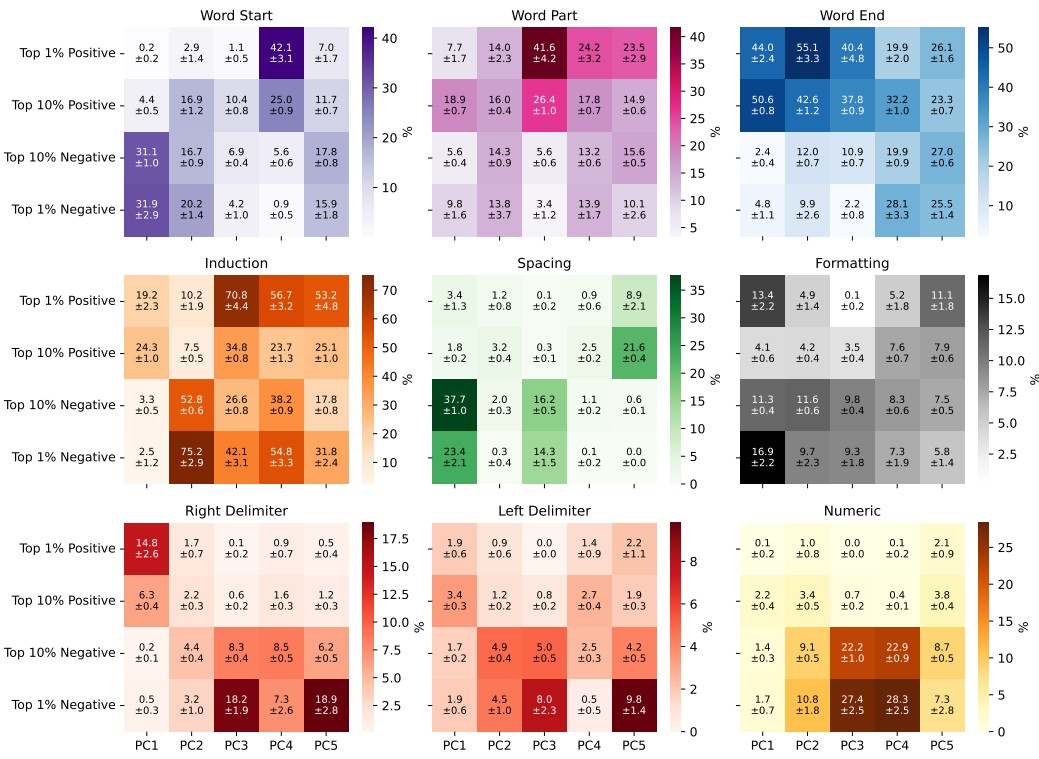

Figure 36: **Per-token susceptibility PCA for layer** 0 **heads** showing mean and standard deviation of loadings of principal components on data patterns across 10 independent draws of 20000 tokens from each dataset.

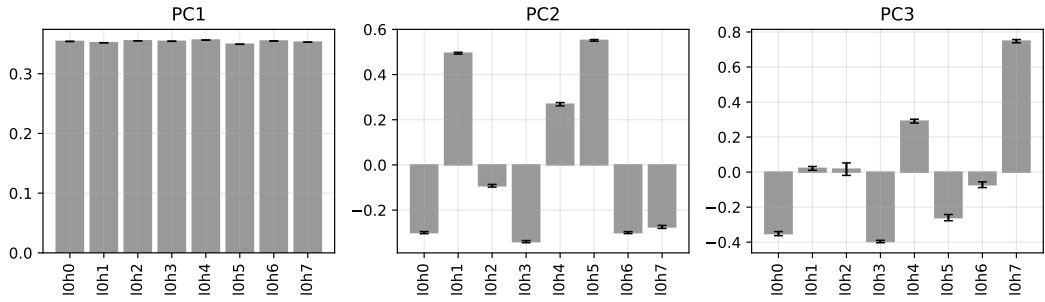

Figure 37: **Loadings of the top three principal components on layer** 0 **attention heads**, showing mean and standard deviation across 10 independent draws of 20000 tokens from each dataset.

# K    STATEMENT ON USE OF LLMS

LLMs were used in the course of this research to support literature review for related work as well as to generate some of the code used to implement experiments and to analyze and plot the data. All LLM-generated output was reviewed by a human author.

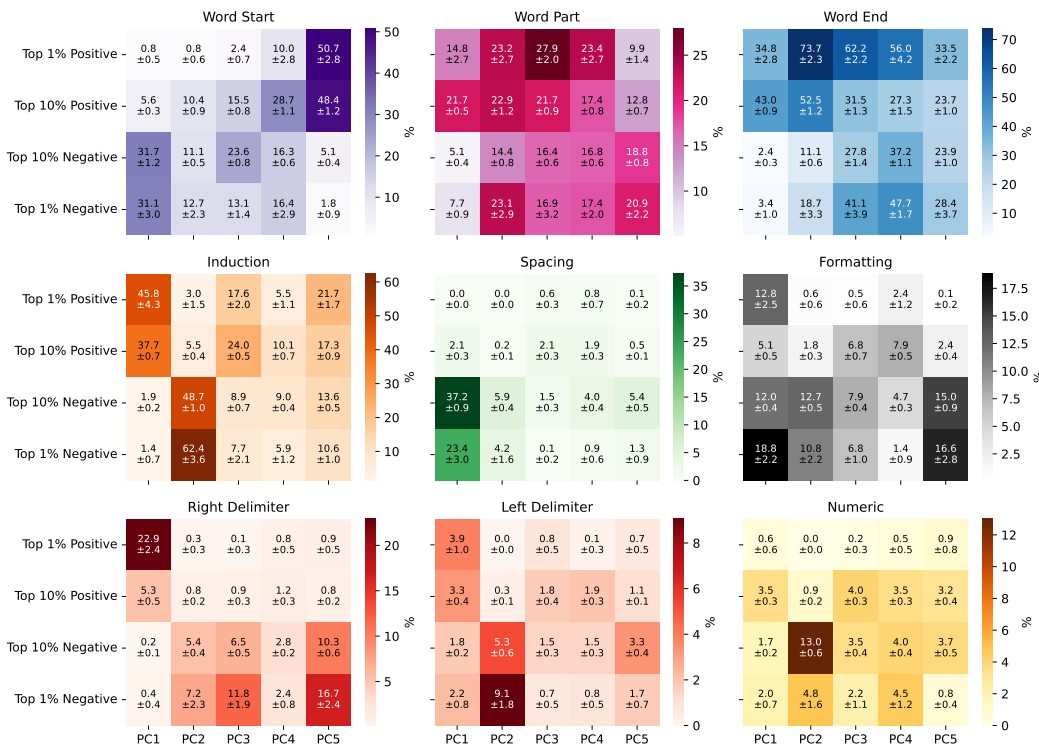

Figure 38: **Per-token susceptibility PCA for layer** 1 **heads** showing mean and standard deviation of loadings of principal components on data patterns across 10 independent draws of 20000 tokens from each dataset.

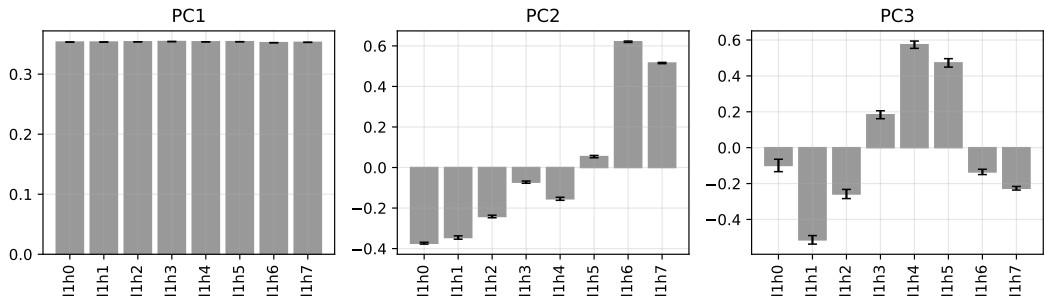

Figure 39: **Loadings of the top three principal components on layer** 1 **attention heads**, showing mean and standard deviation across 10 independent draws of 20000 tokens from each dataset.

