# OpenReview forum: "Structural Inference: Interpreting Small Language Models with Susceptibilities"
_ICLR.cc/2026/Conference — ICLR 2026 Poster_

### Official Review · Reviewer_Wjna · 2025-10-26

**Soundness:** 2
**Presentation:** 3
**Contribution:** 3
**Rating:** 6
**Confidence:** 3

**Summary:**

This paper introduces susceptibilities, a new interpretability framework grounded in Bayesian statistical mechanics that measures how specific neural network components respond to controlled changes in the data distribution. By estimating these susceptibilities locally using SGLD, the authors define a notion of expression vs. suppression for model components and construct response matrices that reveal internal structure. Applied to a 3M-parameter transformer trained on the Pile, the method automatically separates known functional modules such as the induction circuit and identifies linguistic patterns like word segmentation and bracket matching.

**Strengths:**

1. This work introduces a principled and novel framework for understanding the internal computations of deep neural networks, drawing inspiration from Bayesian statistical mechanics and singular learning theory.
2. The proposed methodology offers an unsupervised pipeline for discovering functional circuits within neural networks.
3. It not only provides empirical results demonstrating the effectiveness of the framework, but also includes sanity checks to ensure its practical value in understanding transformer-based language models.
4. The approach establishes strong connections to physics, which may encourage researchers in the field to contribute to the study of the interpretability of neural networks.

**Weaknesses:**

1. Although the theoretical framing is elegant, its practical usefulness remains uncertain. The empirical evaluation focuses only on a small toy model with two attention layers, which is not representative of typical language models.
   - Training on only a subset of the Pile dataset may be too simplistic, potentially explaining why the first principal component accounts for  95–99% of the representational variance.
2. The interpretation of principal components remains largely manual and speculative, which may lead to inconsistent or subjective interpretations of the empirical findings.
3. The proposed susceptibility framework captures correlations in the local loss landscape, in contrast to many mechanistic interpretability approaches that emphasize causal relationships.
4. Finally, the paper requires expertise in Bayesian statistics, which may limit accessibility and reduce its potential audience.

**Questions:**

1. The empirical results in Section 4.2 show that the first principal component (PC1), interpreted as “word segmentation,” accounts for an overwhelming portion of the variance (95.3% in Layer 0 and 99.1% in Layer 1). Could the authors clarify how the proposed method can reveal more complex circuits when the learned representations appear to be dominated by such a simple feature?
2. Since susceptibility is fundamentally correlational, did you conduct any causal validation, such as steering or ablating the susceptibility-identified heads, to assess their influence on the model’s outputs?
3. Appendix C.5 presents a cost model that scales linearly with the number of components. While this may be feasible when analyzing only attention heads, it seems impractical once MLP neurons are included. Do you plan to extend the framework to investigate MLP neurons as well?

---

> ### Author Response · Authors · 2025-11-20
>
> Thanks for your review and the detailed questions and comments.
>
> Regarding scale, please see the top-level comment.
>
> The Pile is a large scale diverse dataset, including Common Crawl which forms the basis of many open-source LLM training datasets. It also includes code and mathematics. In our understanding it is fairly representative of the training distribution of LLMs. While it is true that the dataset is relatively old and there are newer open-source distributions, we prioritised the Pile since we wanted to run follow-up experiments on Pythia models which were trained on this distribution.
>
> Regarding the evaluation of the principal components. We agree there is a manual element to this and that more thorough statistical analysis of this categorisation might strengthen the paper. The categories of tokens were arrived at by studying the loadings of tokens on principal components, and the proportions shown in Fig 2 indicate that the concentrations of high coefficient tokens among these patterns is not subtle. These patterns are also consistent with many other ways of studying language models (e.g. SAE features also show the importance of delimiters and spacing tokens). While there remains an ad hoc element to our choice of token patterns and we cannot claim it is in any sense “canonical” it does seem quite natural.
>
> Regarding correlation vs causation. This is an interesting point and we’re glad to clarify. It is true that many mechanistic interpretability techniques, such as ablations, are causal in the sense that they change the model and observe the effects, and derive from these changes something about internal structure. But this is also the case with susceptibilities, which can be viewed as studying the effect of perturbing weights on model outputs. The *effect* of these perturbations is captured by correlations with respect to the Bayesian posterior, but this is a separate technical matter. In our view susceptibilities are just as causal as ablations, and indeed can be viewed as the “singular learning theory” version of ablations.
>
> Questions:
>
> 1. It is true that PC1 explains a surprisingly large percentage of the variance. But this does not necessarily mean that PC2 and onwards are subtle signals. This is clear for instance in Fig 1\. While it is visible here that the distinction between red and green matches with word ends and word starts and this is the strongest signal, the development of “more green” for induction patterns over the context (see e.g. the “):” token across lines and the end of “sodium”) is also very visible. Since PC2 captures the induction circuit across all four seeds (Appendix I) it seems reasonable to say this signal is robust. It is not clear how robust and meaningful the PCs beyond PC4 are.
> 2. Yes, in effect we did such experiments. The Dyck circuit which appears in PC3 was previously studied in Wang et al 2024 (citation of the paper) and it was observed there that ablating the heads does in fact deteriorate performance on bracket matching. This further matches with the data loadings on PC3, which we view as pretty strong evidence that the analysis by susceptibilities lines up with the analysis via ablations in this case.
> 3. We are currently studying Pythia models, in which we treat MLP layers as a single component (that is, we do not add one component per neuron). This seems to work well and does not expand the component count significantly (indeed most of the count is due to attention heads rather than MLPs). It is true that we would expect finer information the more you decompose components into subcomponents, and more experience will be necessary to know the right balance here.

---

> > ### Comment · Reviewer_Wjna · 2025-11-24
> >
> > I thank the authors for their rebuttal, which provides a helpful clarification of computational scalability and explains how SGLD-based methods could, in principle, extend to larger models with costs comparable to other perturbation-based interpretability techniques. However, this does not fully resolve the deeper concern that the method’s practical scalability and usefulness on realistic LLMs remain unproven: the empirical results rely solely on a very small 3M-parameter model, PCA is overwhelmingly dominated by PC1, the interpretation framework depends heavily on manually defined token categories, and the current approach cannot incorporate MLP neurons in a fine-grained manner, an increasingly important limitation as model complexity grows. Overall, despite its restricted empirical scope, the conceptual contribution remains valuable and supports a weak acceptance.

---

### Official Review · Reviewer_B7pi · 2025-10-29

**Soundness:** 3
**Presentation:** 2
**Contribution:** 3
**Rating:** 6
**Confidence:** 2

**Summary:**

The authors introduce a new interpretability framework grounded in statistical physics and Bayesian learning theory. The key idea is to view a neural network as a Bayesian statistical mechanical system, where the model’s parameters and interactions can be explored through small perturbations of the data distribution. Small shifts in the input data (like moving from natural text to programming code) produce first-order linear responses in specific components of the model. These responses (called susceptibilities) reveal how strongly and in what direction each component reacts to data changes, providing a principled way to quantify its sensitivity and functional role within the network.

**Strengths:**

* The framework is theoretically rigorous, grounded in statistical physics and Bayesian learning theory, while also being empirically validated through concrete experiments.

* The proposed approach is novel:  Susceptibility analysis connects the functional behavior of model components to shifts in the training distribution and shows that heads with similar response patterns cluster into interpretable groups.

**Weaknesses:**

* The analysis (Sec 4) focuses on a very simple set of patterns (Word Start, Word Part, Word End, Induction Pattern, Right Delimiter). In addition, the framework is demonstrated only on a small toy language model (3M, 2 attention layers, without MLP). Although the authors note that they do not anticipate major obstacles in scaling the method to larger models, applying it to a larger model could have strengthened the work by demonstrating the ability to capture more complex behaviors.

* This is only a suggestion, but I think the presentation could be made more accessible to readers who are less familiar with the theoretical background.

**Questions:**

* as I mentioned in the weaknesses section, I think the paper would be stronger if it demonstrated how the proposed method could be applied to analyze more complex patterns or behaviors (such as bias or knowledge acquisition).

---

> ### Author Response · Authors · 2025-11-20
>
> Thanks for the review and questions.
>
> Regarding scaling, we do agree; please see the top level comment. The purpose of the present paper is to thoroughly establish the techniques with a range of sanity checks in small models where we understand more of the ground truth, but we naturally agree the most interesting applications are in larger models.
>
> As we touch on in the top-level comment, more complex patterns will require more complex data analysis. While it is striking that induction patterns and matching brackets appear so strongly (and linearly) in the data in the small model, we expect that some of the more complex and interesting behaviours of language models are not necessarily to be found this way.
>
> The future of the methodology is not to simply move through the principal components in order and find more circuits (although this may be possible in the small model). It is to apply, in addition to PCA, non-linear data analysis methods like UMAP and clustering to the susceptibility data. However, before proceeding to apply more sophisticated techniques, we felt it worthwhile to establish the method in the simplest setting.

---

### Official Review · Reviewer_KJxa · 2025-11-01

**Soundness:** 3
**Presentation:** 2
**Contribution:** 3
**Rating:** 6
**Confidence:** 2

**Summary:**

The authors provide a framework for approximating the effect of a small change in the training data on model components, called susceptibilities. They show that this is equivalent to the covariance between a component’s loss (in this case, attention heads) and the change in total loss under the perturbed data distribution. They then estimate, using locally sampled posteriors with SGLD, susceptibilities for loss on each pair of tokens in the vocabulary, for all attention heads in a 3M-parameter transformer, trained on subsets of Pile. The top principal components of this combined covariance matrix are sensitive to certain classes of interpretable token patterns, such as the induction.

**Strengths:**

1. The assumptions are clearly stated, and the theory seems well grounded.
2. Applying susceptibilities to attention heads yields interpretable patterns, previously found in small attention-only transformers via mechanistic interpretability.

**Weaknesses:**

1. Although novel, this attribution method seems very hard to scale to larger models.
2. The heads that express or suppress the reported patterns in Figure 2 are not yet mechanistically explained. The correlation with Direct Logit Attribution (Figure 32) shows no sign that they are actually responsible for the behaviour.
3. Relaxing sampling to the local posterior makes sense for small changes, but it is unclear if this holds when 10% of the data is replaced. It would be great if this could be clarified.

**Questions:**

1. Could the authors explain why $\delta h = 0.1$ is justified? And additionally, compute $\chi$ for range of values of $\delta h$ to check how different the susceptibilities are?
2.  How to interpret the functional roles of these heads with respect to susceptibilities, given that the DLA plot doesn't show any correlation? One could conduct causal ablations on a few samples as a sanity check, but it is unclear if it would be correlated, given that DLA shows no signs of life. Does this method actually find a causal, induction circuit?

It would be great if the authors could address these two points in the paper (or point me to it, if it already exists).

Some writing suggestions:
1. A notation table (and maybe an algorithm table) would be extremely helpful as the paper is quite mathematically dense.
2. Add some more description about the key findings. Eg: suppression and expression in the introduction

---

> ### Author Response · Authors · 2025-11-20
>
> Thanks for your review, and for offering some points for us to clarify.
>
> Regarding scaling to larger models, please see the top level comment. While the costs do increase with scale, as for many interpretability methods, we are confident the current approach scales at least to 10B.
>
> Regarding mechanistic explanations. It is true that we have no mechanistic explanation for the overall pattern in PC1 and this would be interesting to study further. However the induction circuit (roughly PC2) and Dyck circuit (roughly PC3) were studied mechanistically in this model in Wang et al 2024 (citation from paper) using ablations and other metrics, as described in Section 2.4. We interpret your comment to mean that these mechanistic explanations do not directly connect to expression and suppression?
>
> This is true, and it remains somewhat mysterious how suppression and expression as we describe them using susceptibilities relate to other proposed definitions in the mechanistic interpretability literature such as DLA described in Appendix H. In our view this is an interesting direction for future work. One note we would make is that, while reasonable, the definition of DLA is somewhat ad hoc and based on certain assumptions about how heads influence logits; the lack of correlation with susceptibilities does not preclude that these measurements are correlated with some other more refined mechanistic notion of suppression.
>
> Questions:
>
> Regarding $\delta h$. Please note that this is only relevant for the methodology in Appendix C.3 for the susceptibilities computed for datasets (which appear only in Appendix F and not in the main text) and not for the per-token susceptibilities which are the main objects of study. This was chosen because it was not practical to go lower, with a batch size between 16 and 64 (note that we did test to see that the results in Appendix F were not changed by changing the batch size). We test a range of $\delta h$ values from $0.1$ up to $1.0$. There are only minor differences between $0.1$ and $0.2$, but the difference is substantial (as one would expect) once $\delta h$ increases to $1.0$.
>
> Regarding functional roles. As a conceptual point, we do not view ablations or other current mechanistic techniques as necessarily “ground truth” in which all interpretability claims need to be grounded for them to be valid. However we do agree this is a very useful sanity check\! Fortunately Wang et al 2024 have already characterised the Dyck circuit using just such ablations, and the induction circuit using other standard metrics, to which our results on PC2 and PC3 can be directly compared. So yes, the method does find a causal induction circuit, and circuit for bracket matching.

---

### Official Review · Reviewer_Cmff · 2025-11-03

**Soundness:** 3
**Presentation:** 3
**Contribution:** 3
**Rating:** 8
**Confidence:** 3

**Summary:**

The authors introduce a novel interpretability framework for transformer networks through susceptibilities. The authors investigate how much do variations in the data distribution influence specific components (i.e. attention heads) of the network. The authors highlight two interesting patterns of behavior identified by a standardized notion of susceptibility, resulting from a change in the parameters, namely expression (loss decreases, probability of next token increases) and suppression (loss decreases, probability of next token decreases). Through experiments on a two-layer attention-only Transformer trained on a subset of the Pile, these patterns are related to existing findings from mechanistic interpretability, rediscovering the induction head, as well as identifying how simple linguistic capabilities are conducted within the network.

Overall, the paper presents a strong mathematical foundation and a novel approach for uncovering mechanistic patterns within transformer networks. The paper is well written, albeit the math is quite dense. My main concerns with the paper are the method scaling to larger models, which the authors proactively address in Appendix C.5, but I believe remains a limitation of their work. Finally, I am slightly concerned how the method should be applied in cases where there is no clear next-token point where susceptibility should be estimated (open-ended generation tasks), as the studied patterns are relatively simple to identify from data.

**Strengths:**

- Strong mathematical foundation
- Provides an empirically validated novel method for discovering mechanistic patterns within transformer networks

**Weaknesses:**

- Computational costs when scaling (mentioned as a weakness)
- Applicability beyond simple tasks

**Questions:**

See above

---

> ### Author Response · Authors · 2025-11-20
>
> Thanks for your review and your questions.
>
> Regarding scaling, please see the top level comment.
>
> Regarding tasks where there is “no clear next-token point”. If we understand correctly, the question is: you study how transformers perform on the pre-training next-token prediction task, and show susceptibilities reveal something about some of the patterns in the data and the associated internal structure (e.g. induction patterns and induction circuit), but we actually *use* transformers by rolling out sequences of tokens in an auto-regressive manner, where in each step the transformer is being evaluated on its own outputs, and this is not the setting being studied in the paper. Why should we therefore believe that the results in the paper tell us about how transformers are constructing long contexts?
>
> To this we would say that for short contexts, the transformer is likely to construct sequences of tokens that are close to the pre-training distribution. Therefore if we have a set of contexts $x_i$ and we use the transformer to predict continuations $y_i$ (of some length, but not too long) then evaluating susceptibilities on all token positions in the concatenation $x_iy_i$ should be an appropriate way of studying the structures involved in generating these contexts. If the string generated by the transformer is sufficiently long, maybe this now deviates quite far from the training distribution. That is not *a priori* a problem theoretically and we would still expect the methods to work, but it is true we have not studied this setting and it would be an interesting question for future work.
>
> We’re not sure this answers the question you were asking, please feel free to clarify if not.

---

### Author Response · Authors · 2025-11-20
**Top level comment**

Several reviewers asked about scaling to larger models. This is partly addressed in Appendix C.5 but we include here some more detailed remarks. There are two main obstacles to scaling the results in the paper: (i) the computational cost of SGLD sampling and evaluating losses to compute susceptibilities, and (ii) the complexity of larger models.

Regarding (i) there are two main computational costs in our method: (ia) SGLD sampling and (ib) evaluating forward passes to compute losses across our set of tokens at a given SGLD sample.

(ia) It has already been established that SGLD sampling for local learning coefficient (LLC) estimation can be done at scale, see E. Urdshals et al “Compressibility measures complexity: Minimum description length meets singular learning theory” https://arxiv.org/pdf/2510.12077) which studies Pythia models up to 6.9B (see Figure 3 in loc. cit.).

(ib) Computing susceptibilities is more costly than LLC estimation due to the following factors: the number of model components (each of which has its own posterior that needs separate SGLD sampling) and the number of contexts (each of which requires a forward pass at a given SGLD sample in order to compute losses, to put into the formula for the estimated covariance). Apart from the SGLD sampling this scaling behaviour is similar to e.g. circuit discovery via ablations, which also requires evaluating forward passes for a set of model components that are being ablated.

As models scale the memory requirements mean that these tasks need to be parallelised across multiple GPUs, but this is done in the standard way and is well-supported. Overall there are no theoretical or practical obstacles to scaling to 10B and the costs are not prohibitive (estimates are provided for Pythia 1.4B in Appendix C.5).

Regarding (ii) we agree the difficulties here are more serious. The small transformers studied in the present paper are simple, and so the patterns in the data and corresponding structures in the model are remarkably visible in the susceptibility matrix. We do not expect PCA on its own to be sufficient to discover interesting structure in larger models, which “understand” more complex patterns. In these settings we also make use of non-linear dimension reduction techniques like UMAP, and more sophisticated data analysis to study the internal structure in models. The present paper should be viewed as validation that simple data analysis, in simple models, reveals simple patterns; we agree that it remains an open question (to be addressed in future work) whether more complex data analysis reveals complex structures in larger models.

For intuition, our method should be thought of as somewhat comparable to EK-FAC for data attribution: while it is computationally nontrivial to estimate Hessian inverses in large models and the results are not straightforward to interpret, this is routinely done (see e.g. R. Grosse et al “Studying large language model generalization with influence functions”) because the information gained is believed to be worth the cost.

---

### Meta-Review · Area_Chair_Qd7v · 2026-01-06

**Summary:**

Reviewers agree that the paper is novel and rigorous. Reviewers were generally concerned about scalability, which the authors acknowledge as a weakness.

**Reviewer Concerns:**

Scalability is still a concern. The authors acknowledged it but hope that the insights their framework provides are worth it, and they may be at moderate scales.

**Reviewer Scores:**

I don't see any rebuttal points or new results that would have changed scores. The reviewers might have raised scores out of politeness or to acnowledge the rebuttal effort, but there is no major change. The original reviews were already positive.

---

### Decision · Program_Chairs · 2026-01-26

Accept (Poster)